# Hidden Markov models reveal behavioral state dynamics in depth-related locomotion in mice

**Hironobu Shuto**[1,2,3], **Toshiki Maeda**[2], **Chieko Koike**[4,5,6], **Masayo Takahashi**[1,7], **Michiko Mandai**[8,9], **Take Matsuyama**[8,9,10]*

1 Vision Care Inc., Kobe, Hyogo, Japan, 2 Graduate School of Pharmacy, Ritsumeikan University, Kusatsu, Siga, Japan, 3 Laboratory for Animal Resources and Genetic Engineering, RIKEN Center for Biosystems Dynamics Research, Kobe, Hyogo, Japan, 4 Center for Systems Vision Science, Organization of Science and Technology, Ritsumeikan University, Kusatsu, Shiga, Japan, 5 Ritsumeikan Global Innovation Research Organization(R-GIRO), Ritsumeikan University, Kusatsu, Siga, Japan, 6 College of Pharmaceutical Science, Ritsumeikan University, Kusatsu, Shiga, Japan, 7 Ritsumeikan Advanced Research Academy, Ritsumeikan University, Kusatsu, Shiga, Japan, 8 Research Center, Kobe City Eye Hospital, Kobe, Hyogo, Japan, 9 Research Organization of Science and Technology, Ritsumeikan University, Kusatsu, Shiga, Japan, 10 VCCT Inc., Kobe, Hyogo, Japan

* matsutakehoyo@gmail.com

## Abstract

Understanding how mice process and respond to visual depth cues is crucial for studying visual perception, yet traditional behavioral analyses often miss key aspects of this process, such as the dynamic transitions between behavioral states and the integration of multiple spatial cues that shape depth-related behaviors. Here we demonstrate that mouse responses to visual depth cues are more sophisticated than previously recognized, involving both direct avoidance behaviors and complex modulations of exploratory patterns. By combining a modified circular apparatus with Hidden Markov Model analysis, we reveal that mice transition between three distinct behavioral states—resting, exploring, and navigating—in response to visual depth cues. Using this framework, we uncover several fundamental aspects of mouse visual processing: depth perception has an optimal range of spatial frequencies, with strongest responses to patterns between 6–8 cm; visual processing integrates multiple spatial cues rather than triggering simple avoidance; and initial strong cliff-avoidance responses evolve into more nuanced behavioral adaptations over time. Comparisons between wild-type C57BL/6J mice (*Mus musculus*), retinal degeneration models (rd1-2J, C57BL/6J background, *Mus musculus*), and control conditions confirm that these behavioral patterns specifically reflect visual processing rather than general exploratory behavior. These findings reveal that mouse depth perception involves sophisticated neural processing that modulates overall exploratory behavior rather than simply triggering avoidance responses. Our approach establishes a new framework for analyzing complex behavioral sequences in neuroscience research, demonstrating how refined behavioral analysis can reveal previously undetectable aspects of sensory processing.

**Data availability statement:** The Stan implementations of our models, along with pre-processing and analysis scripts, are available at https://github.com/matsutakehoyo/Hidden-Markov-Model-for-visual-cliff. The repository includes the hierarchical Hidden Markov Model implementation, mixture models for preliminary analyses, and data processing pipelines used in this study. The minimal dataset necessary to replicate the study findings, including raw coordinate data and all scripts used to generate the results, has been deposited in Zenodo and is publicly accessible at: https://doi.org/10.5281/zenodo.15770850.

**Funding:** This study was financially supported by the Japan Society for the Promotion of Science (JSPS) (https://www.jsps.go.jp) in the form of a KAKENHI grant (24H00747) received by TM. No additional external funding was received for this study.

**Competing interests:** Authors with competing interests TM, SH, MT are employees of Vision Care Inc., a startup working on the development of treatments for vision restoration.

## Introduction

Understanding visual function and cognitive behavior in mice is crucial for neuroscience research, as mice serve as a primary model organism for studying the neural mechanisms underlying perception, cognition, and behavior. Reliable assessments of mouse visual function are essential for investigating visual processing and evaluating the effects of genetic manipulations, pharmacological interventions, and neurological diseases on sensory and cognitive functions. While numerous experimental approaches have been developed to evaluate visually-guided behavior in animals (Table 1), many present significant limitations.

Current methodologies broadly fall into several categories (Table 1). Threat-response paradigms, such as looming and sweeping stimuli tests, assess defensive behaviors but often induce stress that may confound results. Prey capture tasks evaluate complex visual tracking but are complicated by prey behavior variability. Learning-based approaches like the Barnes maze and Morris water maze provide insight into spatial memory but require extensive training and can be stressful for animals. Simpler tests such as the optomotor response provide basic measures of visual acuity but fail to capture cognitive processes.

The visual cliff test has emerged as a valuable tool for assessing depth perception and visual function in mice (Table 1). This paradigm leverages animals' innate ability to perceive and respond to depth cues, requiring neither training nor external motivation. However, traditional implementations of the visual cliff test have limitations that may hinder a comprehensive understanding of visually guided behavior. Most notably, the commonly used rectangular apparatus design often leads to corner preference behaviors, which can mask or distort the mice's responses to depth cues (Fig 1A and 1C). Additionally, conventional analysis methods typically reduce complex behavioral sequences to simple metrics such as time spent on either side of the cliff, potentially overlooking important temporal and spatial patterns.

Importantly, animals constantly balance exploratory and defensive behaviors, especially in ambiguous or risky environments. When approaching a potentially threatening stimulus—such as the visually defined cliff—defensive responses tend to dominate. However, with repeated exposures, habituation may occur, shifting the balance back toward exploration. This dynamic interplay between behavioral systems is central to understanding how mice adapt their locomotor strategies in response to depth cues. By examining these state transitions, we can better interpret the observed behavioral signatures and situate them within broader frameworks of behavioral regulation, attention, and adaptation.

Together, these limitations in apparatus design and analysis methods (summarized in Table 1) underscore the need for more nuanced approaches to understanding how mice process and respond to visual depth cues. There is a clear need for methodology that can provide more detailed, quantitative insights into visual behavior while minimizing confounding factors such as stress, learning requirements, and spatial biases.

Current methodologies for assessing visual behavior in mice face three fundamental challenges. First, traditional metrics like time spent in different zones show high variability and are often confounded by non-visual behaviors such as general

**Table 1. Overview of visual function assessment methods in mice.**

| Method | Purpose | Advantages | Limitations | References |
|---|---|---|---|---|
| Looming/ Sweeping Stimuli | Evaluates defensive response to perceived threats | Simple setup; clear behavioral endpoint | Limited behavioral scope; high stress levels | [1–3] |
| Prey Capture | Assesses hunting and tracking behavior in response to prey | Naturalistic behavior; tracks complex movements | Complex analysis; prey behavior may confound results | [4–6] |
| Three-Chamber Social Test | Measures social preference and exploratory behavior | Non-stressful environment; social relevance | Limited to social behavior; does not evaluate visual cognition | [7–9] |
| Barnes Maze | Tests spatial learning and memory based on visual cues | Widely used; well-validated for learning and memory | Learning curve may confound results; animal stress likely | [10] |
| Morris Water Maze | Assesses spatial learning and memory relying on visual cues | High spatial complexity; sensitive to learning and memory | High stress levels; training required; complex data | [11,12] |
| Optomotor Response (OMR) Test | Measures visual acuity and contrast sensitivity | Non-invasive, quantitative measure of visual acuity | Limited to basic visual reflexes, not cognitive responses | [13–18] |
| Visual Discrimination Task | Assesses visual pattern discrimination | Cognitive insights; allows for discrimination learning | Training-intensive; influenced by motivation or reward learning | [19–21] |
| Gap-Crossing Task | Evaluates depth perception | Simple and direct measure of depth perception | Restricted to depth perception without complex visual processing | [22] |
| Visual Water Task | Measures visual discrimination using escape platform cues | Useful for visual learning and memory; quantitative | Stressful; extensive training required | [23–25] |
| Open Field Test with Visual Cues | Assesses exploratory and avoidance behaviors | Simple setup; widely used in anxiety and cognition studies | Limited to anxiety and exploratory behaviors, not specific to vision | [26] |
| Modified Visual Cliff Test (Proposed) | Quantitatively assesses visual cognitive response using HMM | Minimizes stress; provides granular, quantitative data | Novel method; requires HMM modeling | [27–31] |

exploration or anxiety. Second, conventional apparatus designs, particularly square chambers, introduce spatial biases through corner preferences that can mask or distort visually-guided responses. Third, simple analytical approaches that reduce complex behavioral sequences to binary classifications fail to capture the rich temporal and spatial dynamics of visual processing behavior.

These limitations have hindered our ability to fully understand how mice process and respond to visual information. For example, in the visual cliff test, identical measurements of time spent on the shallow side can arise from fundamentally different behavioral patterns – one reflecting true visual processing and another driven by unrelated spatial preferences. Moreover, the high variability in traditional metrics makes it difficult to detect subtle differences in visual processing or to evaluate how responses might change with different visual stimuli or over time.

This table summarizes commonly used methods to evaluate visual function and cognitive behaviors in mice, highlighting their purposes, advantages, and limitations.

To address these challenges, we developed an integrated experimental and analytical framework that combines three key innovations. First, we redesigned the visual cliff apparatus with a circular geometry that eliminates corner preferences and promotes more natural exploratory behavior. Second, we implemented high-precision movement tracking using DeepLabCut, enabling detailed, non-invasive analysis of behavioral dynamics. Third, and most significantly, we developed a novel analytical framework using Hidden Markov Models (HMMs) that can identify distinct behavioral states and characterize how transitions between these states are modulated by visual stimuli. While each of these components builds on existing methods, their integration creates a powerful new approach for studying visual processing behavior. This framework allows us to:

1. Distinguish between visually-guided behaviors and general exploratory patterns

2. Account for and control spatial biases that confound traditional measures

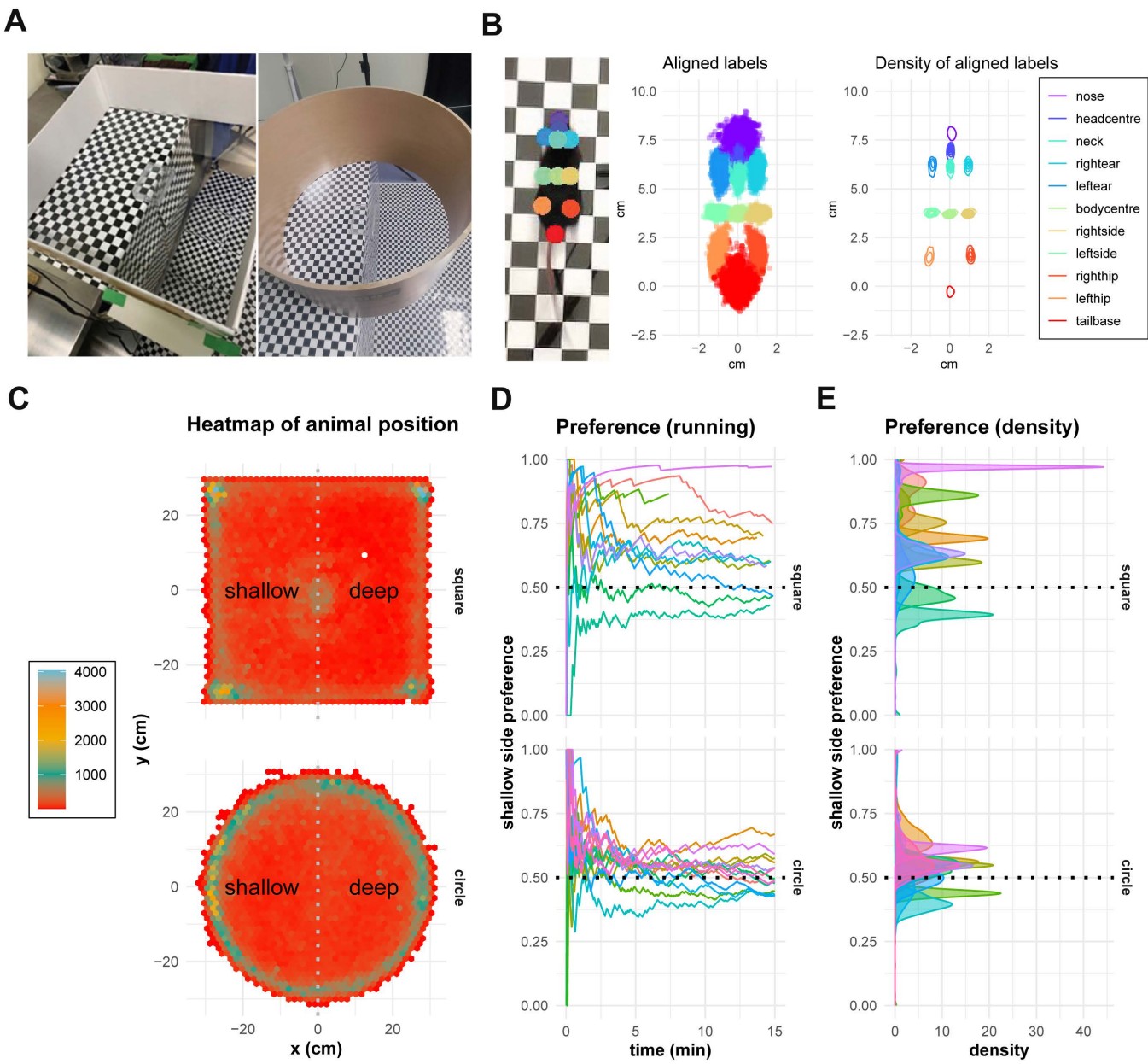

**Fig 1. Behavioral comparisons between square and circular visual cliff apparatuses. (A)** Visual cliff test device (Left: square type, Right: circle type). **(B)** Spatial Distribution of Key Anatomical Labels Captured by DeepLabCut (DLC). The density and spatial distribution of twelve anatomical labels tracked by DLC, representing distinct points across the mouse's body, including nose, head center, ears, hips, and tail base. **(C)** Heatmaps of animal positions in square and circular setups. Both designs show edge preference, but the circular apparatus encourages more consistent engagement with the cliff. **(D)** Running average of shallow-side preference over time. In the square apparatus, mice demonstrate a strong and persistent preference for the shallow side. In the circular apparatus, the shallow-side preference stabilizes near 0.5, indicating more balanced exploration of both the shallow and deep sides. Values >0.5 indicate shallow-side preference, whereas values <0.5 reflect deep-side preference. **(E)** Shallow (table) side preference over time. Variability in the square apparatus (top) contrasts with the consistency achieved in the circular design (bottom), reflecting the reduction of spatial biases in the circular setup.

3. Capture the temporal evolution of behavioral responses to visual stimuli

4. Quantify how specific visual parameters modulate behavior

In our adaptation, we modified the enclosure from the conventional rectangular shape to a circular design, which encourages more natural, unrestricted movement by eliminating corners. In rectangular setups, mice often exhibit a preference for staying in corners, which limits their exploration near the cliff edge and reduces the likelihood of observing behaviors directly influenced by the visual cliff. The circular enclosure overcomes this limitation, allowing for more comprehensive behavioral assessments, particularly near the cliff. Our approach minimizes stress and eliminates the need for learning or training, providing a more accurate reflection of innate visual cognitive behaviors.

One of the key advancements in our approach is the use of machine learning tools to obtain highly precise positional data. Traditional methods of tracking animal movement often involve tagging or manual annotation, which can be invasive or lack precision. We employed DeepLabCut, a tool that leverages deep learning to track body part coordinates with remarkable accuracy (Fig 1B) [32]. This tool enables the collection of large amounts of highly precise, non-invasive data from video recordings without the need for special equipment or invasive tagging. By extracting detailed positional data from the animals' movements, we gain new insights into their behavior in response to visual stimuli.

To analyze the complex behavioral patterns captured by DeepLabCut, we utilized hidden Markov models (HMMs), which allow us to identify distinct behavioral states and characterize how mice transition between these states in response to visual stimuli. While HMMs have been successfully used in ecology to classify animal behaviors and in medical research to track disease progression, their application to visual behavior analysis in controlled laboratory settings remains largely unexplored.

The combination of our circular apparatus design with sophisticated behavioral tracking and HMM analysis allows us to capture subtle behavioral changes, distinguish between different types of movement patterns, and generate statistically robust measures of visual function while accounting for confounding factors such as edge preference and general exploratory behavior. To our knowledge, this represents the first application of HMMs to assess visual cognitive behavior in animals, providing a powerful new tool for understanding visual processing and its impairment in disease models.

The remainder of this paper is organized as follows. We first describe our experimental methodology, including the design rationale for our circular apparatus, movement tracking procedures, and the mathematical framework for our HMM analysis. We then validate our approach using three experimental groups: wild-type C57BL/6J mice (WT, *Mus musculus*), retinal degeneration model mice (RD, rd1-2J C57BL/6J background, *Mus musculus*), and a control group tested in an open field without visual cliff cues (OF). Through this comparison, we demonstrate that our method can reliably detect and characterize visually-guided behaviors. We next explore how varying the properties of visual stimuli—specifically the size and contrast of checkerboard patterns—affects behavioral responses. Finally, we examine the temporal dynamics of visual cliff responses, revealing how behavioral patterns evolve over the course of experimental trials. Our results demonstrate that this integrated approach provides deeper insights into visual processing than traditional methods, and offers a robust framework for quantitative behavioral analysis.

## Materials and methods

### Animals

All animal experiments were approved by the Institutional Animal Care Committees of RIKEN and Ritsumeikan University and were conducted in accordance with the guidelines for the care and use of laboratory animals. All efforts were made to minimize animal suffering and reduce the number of animals used.

Wild-type C57BL/6J mice (WT, *Mus musculus,* 52 males and 52 females) were obtained from Jackson Laboratory via CLEA Japan. rd1-2J (RD, C57BL/6J background, *Mus musculus,* 5 males and 5 females) mice were obtained from the

RIKEN Center for Biosystems Dynamics Research (BDR). Mice were housed in groups of 2–4 per cage under a 12-hour light/dark cycle (light period: 11:00–23:00; dark period: 23:00–11:00) with *ad libitum* access to food and water.

All behavioral tests were conducted at 8–9 weeks of age, between 9:00 and 11:00, which is 2 hours before the end of the dark period. This timing corresponds to the early subjective morning for the mice, a period when they are typically most active due to their crepuscular nature, exhibiting peak activity during dawn and dusk periods [33,34]. Conducting experiments during this period helps to ensure that the mice are naturally alert and engaged in exploratory behaviors, which is important for assessing their visual cognitive functions.

All animals were anesthetized only at the time of sacrifice. Euthanasia was performed following AVMA Guidelines for the Euthanasia of Animals: 2020 Edition. Mice were first anesthetized by intraperitoneal injection of a triple anesthetic mixture consisting of medetomidine hydrochloride (Domitor, 1.0 mg/mL; Nippon Zenyaku Kogyo Co., Ltd.), midazolam (Domilcam Injection 10 mg; Maruishi Pharmaceutical Co., Ltd.), and butorphanol tartrate (Betorfal, 5 mg; Meiji Seika Pharma Co., Ltd.), diluted in physiological saline (Otsuka Pharmaceutical Co., Ltd.). Subsequently, euthanasia was carried out using carbon dioxide inhalation in a dedicated chamber, with gas concentration increased at a displacement rate of 30–70% of the chamber volume per minute. $CO_2$ flow was continued for at least one minute after respiratory arrest to ensure complete euthanasia. These procedures were implemented to minimize distress and pain, including the gradual increase in $CO_2$ concentration and continued flow to reduce the time between labored breathing and loss of the righting reflex.

## Experiment apparatus

A visual cliff was created using an 800 mm high table covered with a black and white checkerboard pattern. A transparent acrylic plate was placed over the checkerboard to create the illusion of a cliff (square apparatus: 650 mm length × 900 mm width × 5 mm thickness; circular apparatus: 1000 mm length × 1000 mm width × 5 mm thickness).

In the square apparatus, a wooden enclosure (homemade, covered with styrene sheeting) measuring 600 mm in length and width and 150 mm in height was positioned to straddle the boundary of the cliff. A transparent platform (110 mm long × 80 mm wide × 30 mm high) was placed at the center of the apparatus as the starting point for each mouse. The checkerboard pattern used in the square apparatus had a square size of 23.5 mm with a black-to-white contrast ratio of 1:0 (maximum contrast).

In the circular apparatus, a circular enclosure made of a paper tube (KOMETANI PAPER TUBE Mfg CO.,LTD.) with an inner diameter of 600 mm and a height of 360 mm was placed on the acrylic plate, straddling the cliff boundary. This circular design was chosen to eliminate corner preference and encourage natural exploration. A transparent platform (110 mm long × 80 mm wide × 30 mm high) was placed at the center of the apparatus to serve as the starting point for each mouse. For the circular apparatus, we used black and white checkerboard patterns with multiple square sizes (1, 5, 20, 40, 60, 80 mm) and various black/white contrast ratios (1:0, 1:0.25, 1:0.50, 1:0.75, 1:1, 0:0.25, 0:0.50, 0:0.75).

Lighting conditions were carefully controlled in both setups. In the square apparatus, uniform illumination was provided by two warm white light bulbs (peak wavelength 585 nm, 16 cd/m², LDA6L/3 TOSHIBA) placed at a height of 1900 mm to maintain consistent brightness. In the circular apparatus, overhead lighting devices (3000 K, peak wavelength 650 nm, 65 cd/m², IVISII) were mounted at the same height (1900 mm) to provide uniform illumination. These lighting devices allowed adjustment of both color temperature and intensity. An open field test was conducted across different color temperatures (3000K, 6500K, 8000K) and light intensities (1%, 50%, 100%) to determine optimal conditions. No significant differences in locomotor activity were observed across these conditions, though consistent with prior reports [33,34], mice are generally more active at dawn/dusk light levels and exhibit reduced anxiety-like behaviors under warm, low-intensity lighting [36]. Therefore, we selected 3000K (close to dawn/dusk color) at 50% intensity to balance visibility for DeepLabCut analysis with naturalistic behavior, avoiding 100% intensity, which was deemed too bright.

Preliminary trials were conducted using both square and circular arena setups under varying lighting conditions to assess the potential influence of lighting, floor reflections, and subtle visual asymmetries on locomotor behavior. We

consistently observed corner-staying behavior in the square arena regardless of lighting conditions, while the circular arena consistently promoted more uniform spatial exploration. Although the square and circular setups were not strictly matched for lighting or other visual features, the robust reduction in corner-staying behavior in the circular arena was consistent across these different conditions, suggesting that the arena shape was the primary driver of the observed behavioral differences.

## Experiment procedure

At the start of each trial, a mouse was placed on the central platform within the apparatus, facing the shallow side of the cliff. In this study, the starting directions of all mice were made the same in order to keep non-visual conditions as consistent as possible. Mice were allowed to voluntarily disembark from the platform and explore the apparatus freely for approximately 10 minutes. If a mouse did not leave the platform within 5 minutes of the start of the experiment, it was gently guided off the platform facing the shallow side. The apparatus was uniformly illuminated using the overhead lighting described above. Behavioral activity was recorded from above using a camera (SANYO, recording at 30 frames per second) for subsequent analysis.

## Mouse body position tracking

All experimental videos were analyzed using DeepLabCut [32], an open-source software for pose estimation and tracking using deep learning. Twelve key points on the mouse's body were manually labeled on randomly selected frames to train the network for accurate tracking [37]. These key points included the nose, head center, left and right ears, neck, body center, left and right sides, left and right hips, and tail base. These 12 points were frequently labeled in DeepLabcut's analysis and helped to properly capture the mouse's movements. In this study, we calculated the movement metrics using the body center coordinate, which has the highest confidence among all parameters. From the DeepLabCut output, the x and y coordinates of the body center were extracted for each analyzed frame. Frames where the tracking confidence was below a 90% threshold were excluded from the analysis to ensure data quality. Frames below 90% contain body parts that are not labeled, which poses a risk of not being properly analyzed, so we chose frames with a confidence of 90% or higher.

## Coordinate system adjustment

To standardize positional data across trials and ensure consistent analysis of mouse movements relative to the cliff boundary, we adjusted the coordinate system for each trial using the recorded positional data. Variations in camera placement and slight shifts in the apparatus between trials necessitated the estimation of the center and orientation of the circular enclosure for each experiment.

First, we estimated the center $(x_c,\ y_c)$ and radius $r$ of the circular apparatus for each trial by fitting a circle to the distribution of the mouse's body center coordinates. This was achieved using an iterative grid search algorithm that maximized the proportion of positional data points lying within the candidate circle. The algorithm refined the estimates until convergence criteria were met, ensuring accurate representation of the enclosure's dimensions. With the estimated center and radius, we translated the coordinate system so that the center of the apparatus corresponded to the origin $(0,0)$, standardizing the positional data across all trials and allowing direct comparison of mouse movements relative to the enclosure's center.

To account for any rotational misalignment of the cliff boundary due to camera angle, we aligned the cliff boundary vertically along the $x = 0$ axis. Two points along the cliff boundary were manually identified from video frames for each trial, providing the coordinates necessary to define the cliff boundary line. The angle $\theta_{Cliff}$ between the cliff boundary and the vertical axis was calculated using the arctangent of the slope formed by the two identified points. We then rotated the positional data by angle $-\theta_{Cliff}$ using a rotation matrix, aligning the cliff boundary vertically.

After translation and rotation, the coordinate system was standardized: the cliff boundary corresponds to $x = 0$, effectively dividing the apparatus into two regions; the shallow side (table side) is represented by, $x < 0$, located to the left of the origin; the deep side (cliff side) is represented by $x > 0$, located to the right of the origin; and the $y$-axis remains the vertical axis of the apparatus, with values increasing toward the top. This standardized coordinate system ensured that all positional data were aligned relative to the cliff boundary, facilitating consistent and accurate comparisons of movements toward and away from the cliff across all trials. By implementing these adjustments, we ensured that subsequent analyses, including the application of the Hidden Markov Model, were based on accurately aligned and comparable spatial data, which was crucial for the reliable interpretation of the mice's exploratory behaviors in response to the visual cliff.

### Length and angle

From the positional data, two primary movement metrics were derived using the body center coordinates of the mouse. The first metric, step length ($l_t$), represents the distance traveled by the mouse during a defined time interval and is calculated as:

$$l_t = \sqrt{(x_t - x_{t-1})^2 + (y_t - y_{t-1})^2}$$

Where ($x_t$, $y_t$) and ($x_{t-1}$, $y_{t-2}$) represent the spatial coordinates of the mouse at consecutive time points.

The second metric, angle ($\theta_t$), describes the direction of movement during the same interval. This is computed as:

$$\theta_t = tan^{-1}\left(\frac{y_t - y_{t-1}}{x_t - x_{t-1}}\right)$$

These metrics provide quantitative measures of the mouse's activity levels and directional movement, forming the basis for analyzing behavioral patterns and state transitions in the Hidden Markov Model framework.

### Statistical analysis

All statistical analyses were performed using R (version 4.3.1) and Stan (version 2.32.6). We employed Bayesian inference methods for model fitting and validation, which provided several advantages for our study.

We used a Bayesian Hidden Markov Model (HMM) to analyze locomotor behavior in the circular visual cliff arena. This approach was selected a priori because it captures the probabilistic switching between behavioral states and accounts for the sequential nature of locomotor data. The Bayesian framework allows us to incorporate reasonable prior knowledge about environmental factors (e.g., distance to the edge or center) that influence movement. By integrating these covariates and setting informative priors, we aimed to create a flexible model that reflects known behavioral tendencies while allowing for the detection of unexpected patterns. Although other unsupervised methods (e.g., k-means clustering, PCA-based trajectory analysis) exist, we chose the HMM approach because of its superior ability to handle temporal dependencies and interpretability in terms of state transitions.

One of the key benefits of using Bayesian methods is the ability to obtain full posterior distributions of model parameters. These posterior distributions allowed us to quantify the uncertainty associated with parameter estimates and to assess the effects more comprehensively. By analyzing the posterior distributions, we could provide probabilistic interpretations of behavioral states and state transitions, enhancing the robustness and interpretability of our findings.

Detailed statistical analyses, including model specifications, selection of priors, convergence diagnostics, and validation procedures, are presented in the Results section. There, we discuss the selection of HMM parameters and the impact of various modeling decisions on the interpretation of the behavioral data.

## Results

### Effectiveness of the circular apparatus

Traditionally, the visual cliff test has been conducted in a square apparatus, where a pseudo-cliff is created by overlaying a checkerboard pattern on a surface with a transparent acrylic sheet extending over the edge. This setup allows for the observation of depth perception by assessing the proportion of time mice spend on the "table" (shallow) side versus the "cliff" (deep) side. However, one limitation of the square design is that mice often exhibit a strong preference for corners, where they feel more secure. This cornering behavior reduces the likelihood of exploring near the cliff edge, potentially confounding results [38].

In the original human visual cliff experiment [35], parents encouraged infants to cross the visual cliff by calling them from the opposing side, facilitating exploration. In animal models, spontaneous exploration is key, making it essential to design an environment that naturally encourages movement. To address these limitations, we developed a circular apparatus to eliminate corner preference and promote uniform exploration (Fig 1A). The circular design ensures that when mice follow the edge, they inevitably encounter the cliff boundary. Furthermore, its uniform distance from the center to the edge removes directional biases present in square setups.

To evaluate the impact of the apparatus design, we recorded and analyzed the movement trajectories of mice in both the square and circular setups. Mice tended to show a strong preference for edges in both setups (Fig 1C). In the square apparatus, mice often stagnate in corners, while in the circular apparatus, they explore the perimeter more uniformly. The heatmap of positions clearly shows a preference for corners in the square setup. These patterns suggest that the circular design encourages more consistent interaction with the cliff edge.

To further quantify the impact of apparatus design, we analyzed the running average of shallow side preference over time (Fig 1D,1E). In the square apparatus, preference values varied widely across individuals (ranging from 0.5 to >0.75), likely reflecting confounding factors such as corner preferences and uneven spatial exploration. In contrast, mice in the circular apparatus showed more consistent exploratory behavior, with preference values closer to 0.5. However, this more balanced exploration highlighted a fundamental limitation of using time-spent metrics: similar preference values could arise from either random exploration or deliberate interaction with both regions.

These findings highlight the advantages of the circular design in promoting more uniform spatial exploration by eliminating corner preferences and distance biases, addressing key limitations of the traditional square design. By fostering consistent exploration and encouraging greater interaction with the cliff, the circular setup provides a more effective framework for evaluating depth perception and visually guided behavior in mice. However, these results also reveal the necessity for a more nuanced metric beyond relative time spent on the shallow side. The more complex movement patterns we observed demanded a more sophisticated analytical approach that could capture the temporal and spatial dynamics of mouse behavior to fully evaluate the impact of the cliff. This led us to develop a Hidden Markov Model framework capable of identifying discrete behavioral states and characterizing their transitions in response to visual stimuli.

### Hidden Markov models reveal complex behavioral states

Hidden Markov Models (HMMs) provide a powerful framework for analyzing complex behavioral sequences by identifying underlying patterns or "states" that may not be directly observable. These models assume that observable behaviors (like movement patterns) are generated by unobservable states (like different behavioral modes), and that transitions between these states follow probabilistic rules that can be influenced by environmental factors [39]. This framework is particularly well-suited for analyzing mouse behavior in the visual cliff test, where we expect animals to transition between different behavioral states (such as exploration, assessment, and avoidance) based on their perception of depth cues.

HMMs have been successfully applied across diverse fields. In ecology, they classify behaviors of various species into distinct states for detailed analysis [40–42]. In medicine, HMMs predict disease progression and patient outcomes based

on historical data [43], and in sociology, they provide insights into traffic accident patterns [44]. While some studies have used HMMs to classify choice behavior in mice or to examine fly behavior in response to odors [45,46], their application to visual behavior analysis in controlled laboratory settings remains largely unexplored.

While the circular apparatus addresses spatial biases in the visual cliff test, our initial analysis revealed that traditional metrics like shallow side preference fail to capture important aspects of mouse behavior. Even in cases where mice show similar overall spatial preferences, their underlying behavioral patterns can differ dramatically. To illustrate this, Fig 2A shows two distinct movement patterns that would yield identical shallow side preferences (~0.5) but represent fundamentally different behavioral responses to the visual cliff. In one case (Fig 2 left), the animal shows clear state transitions influenced by cliff proximity, alternating between short, random movements (red:Resting) when near the cliff and longer, directed movements (blue:Exploring) elsewhere. In contrast, the other pattern (Fig 2 right) shows high directional persistence in an edge-following behavior, with state transitions occurring independently of cliff position. Traditional metrics would incorrectly classify these distinct behavioral patterns as equivalent responses.

To capture these behavioral differences, we developed a Hidden Markov Model (HMM) framework that identifies discrete behavioral states and characterizes how transitions between states are modulated by environmental features. The HMM framework (Fig 2B) relates observed movement metrics (step lengths $l$ and turning angles $\theta$) to underlying behavioral states that evolve as a first-order Markov chain. Each state is characterized by distinct movement patterns: "Resting" states show short, random steps, while "Exploring" states exhibit longer, more directed movements. To ensure biological plausibility, we implemented a structured transition matrix (Fig 2C) that only allows transitions between adjacent states, reflecting the observation that mice typically change their activity levels gradually rather than abruptly.

A key innovation of our approach is the explicit modeling of how environmental features influence behavioral transitions. The model incorporates three primary spatial features (Fig 2D): distance to the cliff ($d_{cliff}$), distance to the edge ($d_{edge}$), and distance to the center ($d_{center}$). These distances are transformed into influence functions that modulate both the probability of transitioning between states and the characteristics of movement within each state. These spatial features were transformed into continuous influence functions, allowing us to capture how mice modify their behavior in response to their position within the apparatus. Particularly important was the finding that mice respond not just to the cliff itself, but to the complex interaction between all three spatial features.

## Model development and validation

We developed a three-state Hidden Markov Model to characterize mouse behavior in the visual cliff test (S2 Fig). Through systematic analysis, we determined that mouse behavior could be best described by three distinct states: "Resting" (characterized by short, random movements), "Exploring" (intermediate movements with moderate directional persistence), and "Navigating" (longer, directed movements). This three-state model provided the optimal balance between statistical power and biological interpretability. These models incorporate three key environmental influences that shape mouse behavior: distance to the cliff, distance to the edge, and distance to the center. These spatial features were transformed into continuous influence functions, allowing us to capture how mice modify their behavior in response to their position within the apparatus. Particularly important was the finding that mice respond not just to the cliff itself, but to the complex interaction between all three spatial features.

Detailed descriptions of model specifications, parameter choices, and validation procedures are provided in the supplementary materials. The complete implementation, including all analysis code and Stan models, is publicly available at https://github.com/matsutakehoyo/Hidden-Markov-Model-for-visual-cliff.

## Validation of the HMM framework using wild-type, retinal degeneration, and open field groups

To evaluate our HMM framework's ability to capture complex behavioral patterns, we analyzed data from three experimental groups that should exhibit distinct responses to visual depth cues. The wild-type (WT) group was tested in the standard

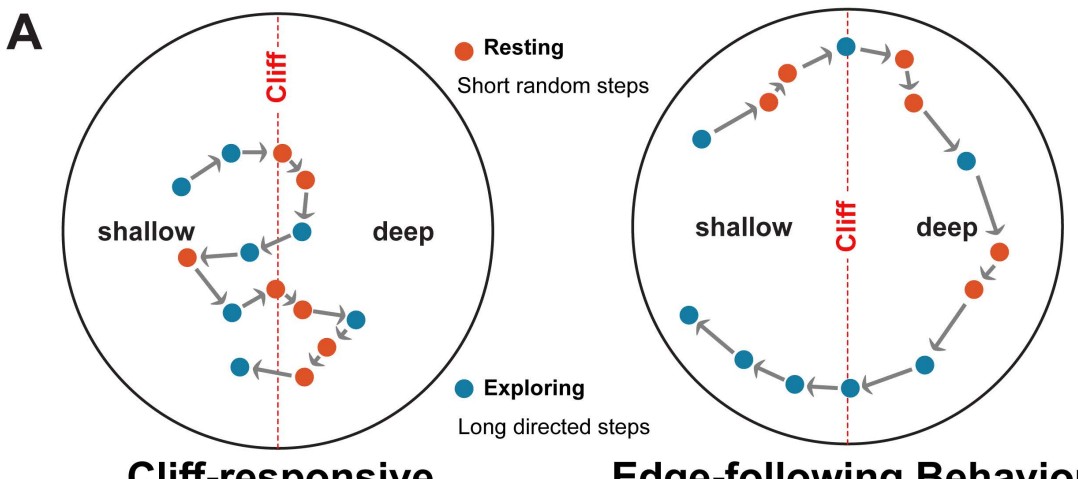

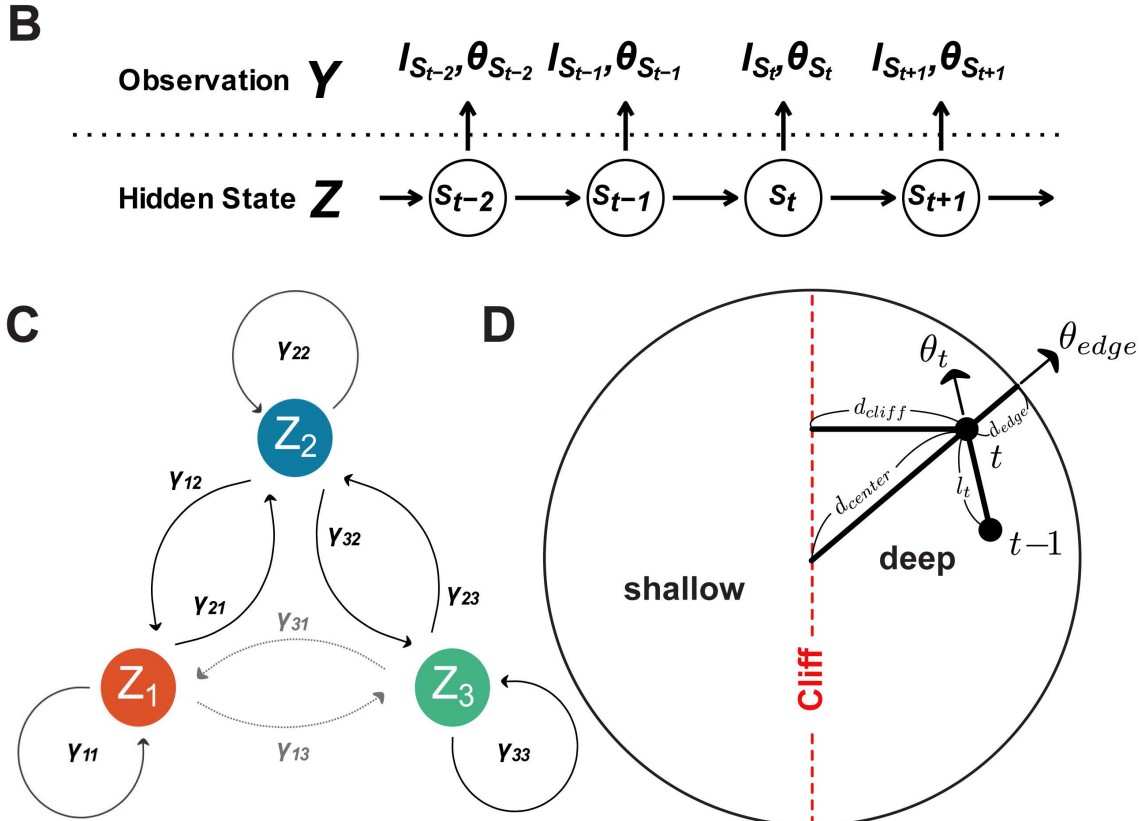

**Fig 2. Hidden Markov Model (HMM) framework for analyzing mouse movement data. (A)** Two contrasting movement patterns with identical shallow side preference (~0.5). Left: Cliff-responsive behavior shows state transitions influenced by cliff proximity and varied movement directions. Right: Edge-following behavior exhibits high directional persistence and state transitions independent of cliff position. States are classified as Resting (red,

short random steps) or Exploring (blue, long directed steps). Arrows indicate movement direction. **(B)** HMM structure showing relationship between observations **Y** (step lengths $l_t$ and turning angle $\theta_t$) and hidden behavioral states **Z**. States evolve as a first-order Markov chain, with each observation depending only on the current state. **(C)** State transition diagram showing allowed transitions between three behavioral states. $\gamma_{ij}$ represents transition probability from state to state **j**. Transitions between adjacent states (**1** ↔ **2** and **2** ↔ **3**) are allowed, while direct transitions between non-adjacent states (**1** ↔ **3**) are restricted, as represented by the grayed-out transitions. **(D)** Spatial features used to model environmental influences on behavior. At each time point t, distances to cliff (**d**$_{cliff}$), edge (**d**$_{edge}$), and center (**d**$_{center}$) are calculated. $\theta_t$ represents the current movement angle, and $\theta_{edge}$ the angle tangent to the circular boundary.

visual cliff apparatus, providing a baseline for normal visual depth perception. The retinal degeneration (RD) group, consisting of rd1-2J mice with impaired visual function, served as a negative control lacking depth perception capability. Additionally, we tested WT mice in an open field (OF) apparatus without depth cues as a second control condition. This design allows us to distinguish visually-guided behaviors from general exploratory patterns.

First, we assessed the model's ability to capture movement dynamics at both individual and group levels. Fig 3A-3B shows posterior predictive checks of step length and turning angle distributions for individual animals across all three groups. The close alignment between observed data (green histograms) and model predictions (purple lines) demonstrates that our framework successfully captures the distinct movement patterns of individual mice. Importantly, the model also accurately reproduces temporal dependencies in movement, as shown by the autocorrelation functions for both step length and turning angle (Fig 3C-3D). The narrow credible intervals (purple shading) indicate robust parameter estimation despite individual variability.

Having validated our model's ability to capture movement dynamics, we next characterized how visual depth perception influences behavioral states and their transitions. Fig 3E shows the predicted step length distributions for each behavioral state across groups. All groups exhibited three distinct movement scales: a "Resting" state with short steps (~0.3 cm, CI$_{89\%}$: 0.3–0.4), an "Exploring" state with intermediate steps (~1.2 cm, CI$_{89\%}$: 0.9–1.3), and a "Navigating" state with longer steps (~2.2 cm, CI$_{89\%}$: 2.0–2.6). The consistency of these distributions across groups indicates that the basic movement repertoire remains intact regardless of visual capability.

However, the angular distributions (Fig 3F) reveal that WT mice show distinct directional patterns near the cliff, with reduced angular concentration suggesting more cautious, varied movements. In contrast, RD and OF groups maintain similar angular distributions across all spatial regions, indicating no specific response to the cliff boundary. These findings demonstrate that visual depth perception in mice manifests not through simple avoidance behaviors, but through subtle modulation of exploratory patterns.

The spatial organization of behavior is further revealed through feature influence profiles (Fig 3G). The edge influence exhibits a sharp increase as the mouse approaches the edge. The range of edge influence is slightly larger in the OF group (about 4.4 (CI$_{89\%}$: 4.0–4.9) cm) compared to WT (3.2 (CI$_{89\%}$: 3.0–3.6) cm) and RD (3.0 (CI$_{89\%}$: 2.7–3.4) cm). This difference may be attributed to variations in stress and comfort levels among the groups. In the OF setup, the absence of depth cues likely creates a less stressful environment, allowing the animals to move more freely and exhibit a broader sensitivity to the edge. In contrast, WT mice, which perceive the cliff, may experience heightened caution due to the visual depth cue. RD mice, which lack visual input and therefore do not perceive the cliff, may display inherently more cautious behavior, resulting in a narrower edge influence range similar to that of WT mice.

### Effects of visual stimulus parameters

We next examined the effects of visual stimulus parameters (Fig 4A,4B). These illustrate the stationary state distributions for WT, RD, and OF groups under two simulated movement paths: horizontal (Fig 4A) and along the edge (Fig 4B). These simulations are designed to assess the influence of spatial features (edge, center, and cliff) on behavioral state distributions. The horizontal path, which is influenced by the edge, center, and cliff features, shows distinct behavioral changes

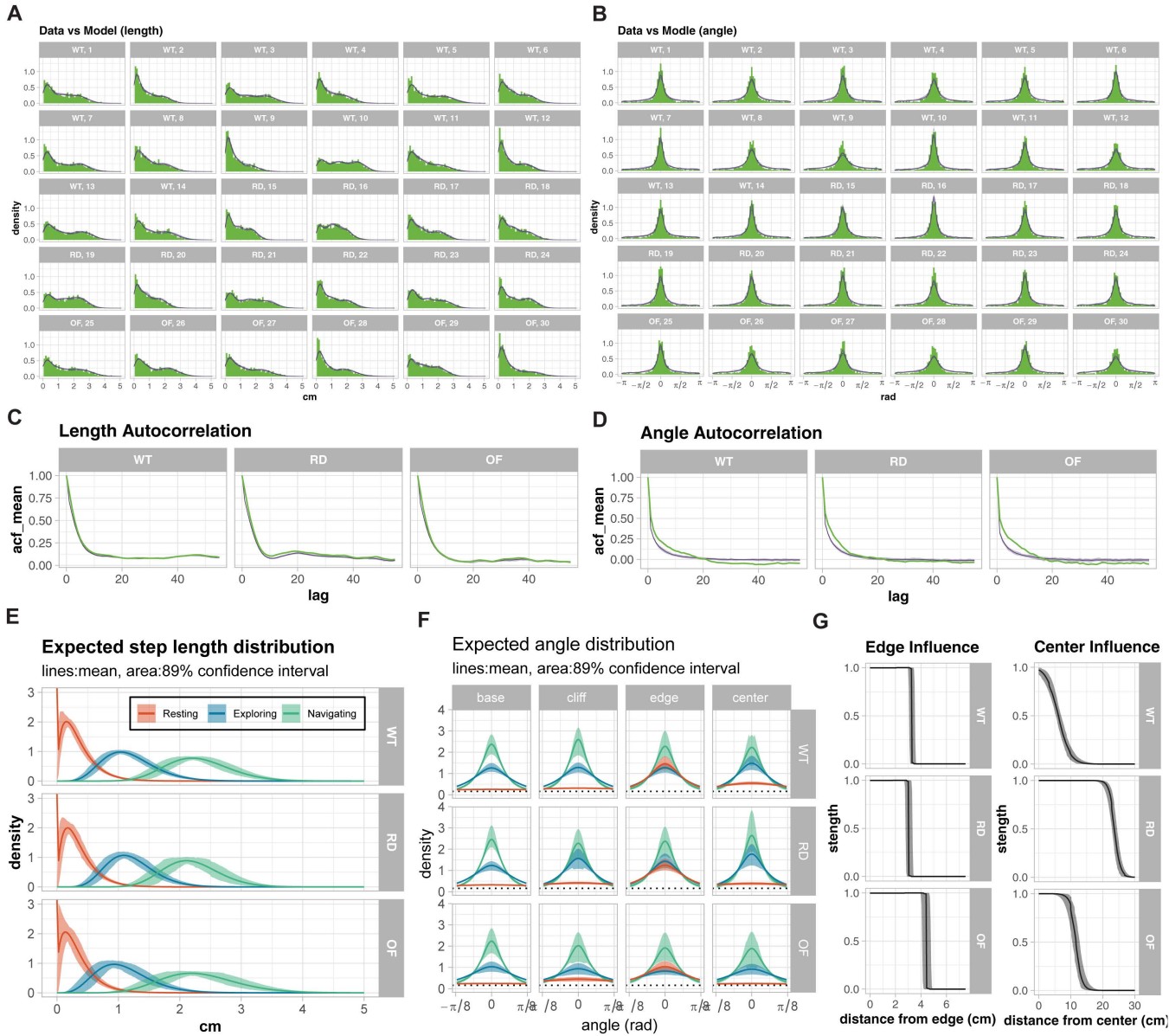

**Fig 3. Validation of the Hidden Markov Model: WT vs RD vs OF. (A-B)** Posterior predictive checks for step length (A) and turning angle differences ($\theta_t - \theta_{t-1}$, B) distributions. Individual facets show data from single animals across wild-type (WT), retinal degeneration (RD), and open field (OF) groups. Green histograms show observed data; purple lines show 10 random draws from model posterior predictions. The close alignment between observations and predictions demonstrates the model's ability to capture movement patterns at the individual level. **(C-D)** Autocorrelation functions for step length (C) and turning angle (D) across experimental groups. Green lines show observed autocorrelations; purple lines and shading represent model predictions and 89% credible intervals. The tight credible intervals and close match to observations indicate the model accurately captures temporal dependencies in movement patterns. **(E)** Predicted Step Length Distributions: Predicted distributions of step lengths for the three behavioral states ("Resting", red; "Exploring", blue; "Navigating", green) in WT, RD, and OF conditions. Solid lines indicate mean predictions, and shaded regions indicate 89% confidence intervals. These distributions highlight consistent state definitions across experimental groups. **(F)** Predicted Angular Distributions: Predicted angular distributions for each behavioral state, shown for basal conditions (leftmost column) and near environmental features: cliff, edge, and center (subsequent columns). The range has been restricted to $[-\pi/8, \pi/8]$ to emphasize differences in angular concentration, with a dashed line marking the uniform random angle density ($y = 0.159$). State 1 ("Resting") exhibits broad angular distributions indicative of undirected movement, while States 2 and 3 show progressively narrower distributions, representing directed movement. Near the edge, angular distributions are sharply constrained, consistent with physical barriers at the enclosure's perimeter. **(G)** Feature Influence profile: Influence of spatial features on movement behavior. (Left) Edge influence as a function of distance from the edge. (Right) Center influence as a function of distance from the center. Lines represent mean predictions, while shaded regions represent 89% confidence intervals. WT, RD, and OF groups exhibit similar edge influence thresholds (~3 cm), while center influence differs markedly, with WT exhibiting the most localized effect and OF the broadest.

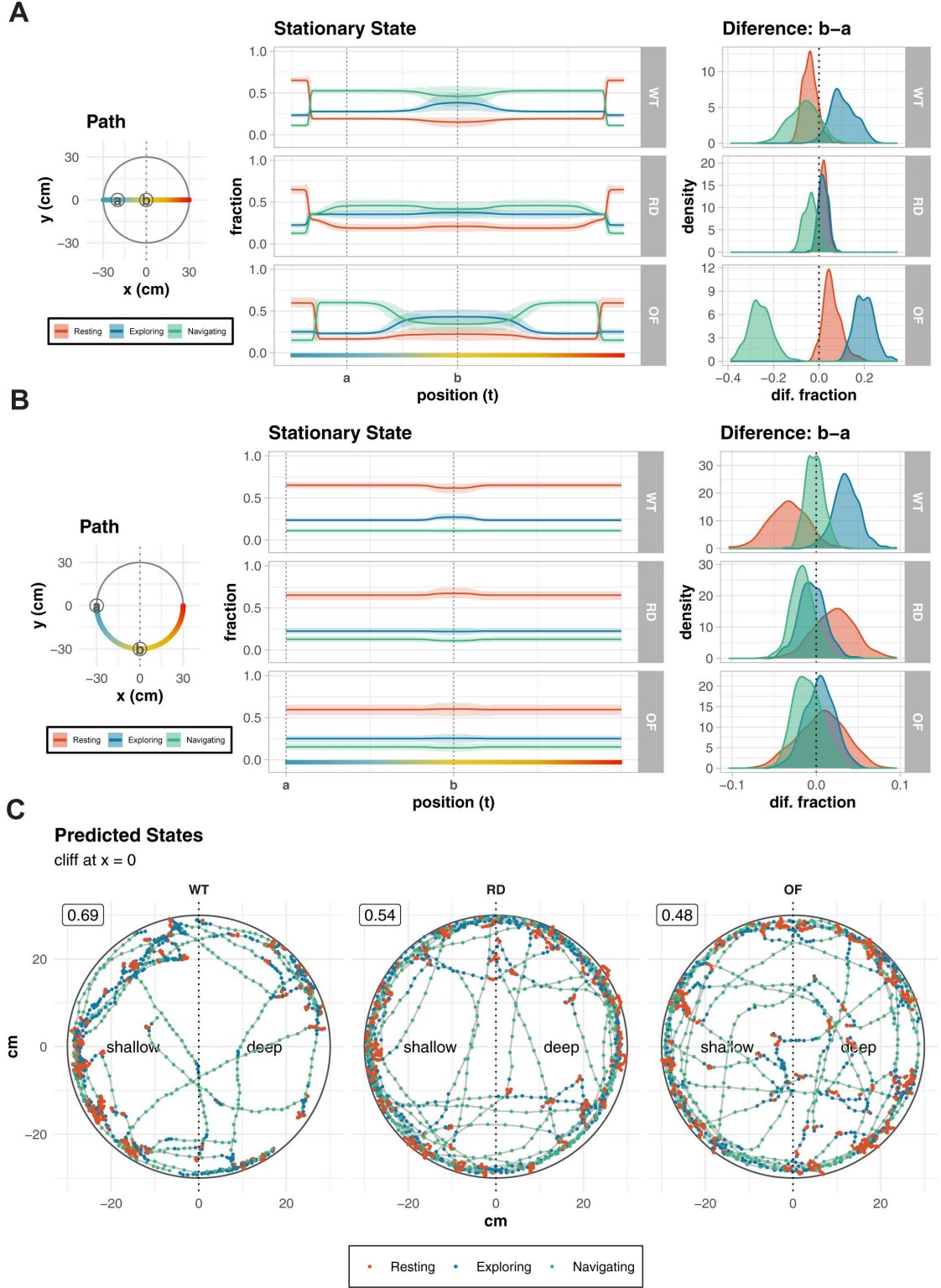

**Fig 4. Behavioral analysis of WT, RD and OF mice. (A-B)** Predicted Stationary State Distributions for Simulated Movements: (Left) Diagrams of simulated movement paths: horizontal (A) and circular paths **(B)**. Key positions "a" and "b" are marked for comparison. The circular path follows the apparatus's outer edge (radius = 30 cm), dominated by edge influence. (Middle) Predicted stationary state distributions along each path, showing the

relative prevalence of "Resting" (red), "Exploring" (blue), and "Navigating" (green) states. (Right) Posterior distributions of state differences between points "a" and "b" quantifying changes in behavior along the path. **(C)** Representative movement trajectories with predicted behavioral states. Points indicate animal position colored by predicted state (red: Resting, blue: Exploring, green: Navigating). Numbers in boxes show shallow side preference rates. Dotted vertical lines mark the cliff boundary ($x=0$), dividing shallow and deep sides. WT mice show structured exploration with clear responses to the cliff, while RD and OF groups display more uniform movement patterns, demonstrating the model's ability to detect behaviorally relevant differences between groups.

near the center and cliff. The circular paths provide a clear view of how behavioral states shift as animals approach the cliff while edge and cliff influences remain constant.

The leftmost panel depicts the movement paths within the apparatus. For the horizontal paths, movement transitions through areas of varying proximity to the center and edge, allowing an evaluation of how spatial features impact state distributions. For the edge path, movement occurs at a fixed distance from both the edge and center, isolating the influence of the cliff.

The middle panel shows the stationary state distributions along the simulated paths. In Fig 4A, near the edge, the edge influence dominates, and "Resting" state is more prevalent. In WT mice, "Resting" declines quickly, and "Exploring" and "Navigating" becomes more prominent as the distance from the edge increases. RD mice exhibit little change in state distributions after moving away from the edge, consistent with blind, exploratory behavior. OF mice (mice experimented in the open field test) show an increase in "Navigating" as they approach the center, indicating more directed exploration. WT mice exhibit a similar but more spatially localized pattern, with "Navigating" increasing only in a restricted central region. In Fig 4B, "Resting" is more prevalent due to the edge effect. In WT mice, "Resting" decreases slightly and "Navigating" increases as the mouse approaches the cliff. Little change is observed in RD and OF mice.

The rightmost panels display the difference between two key points ("a" and "b") on the simulated paths, as indicated in the path diagrams and the stationary state plots. These differences are represented as posterior distributions of the change in fraction for each state. In Fig 4A, For the WT group, a difference of about 10% was observed in "Exploring" compared to the RD group. In the OF group, because of the center effect, "Exploring" increased by about 20% and "Navigating" decreased by about 25%. In Fig 4B, for the WT group, significant differences (~4%) are observed for "Resting" and "Navigating", reflecting a measurable influence of the cliff on stationary state distributions. This effect is absent in both RD and OF groups.

The model's ability to identify meaningful behavioral states is also illustrated in Fig 4C, where representative trajectories are color-coded by predicted state. Compared to RD and OF groups, WT mice show structured exploration with state transitions near the cliff boundary. RD and OF groups but not WT show a higher proportion of "Resting" near the cliff. While all groups show a proportion of shallow side preference (numbers in boxes), state-based analysis reveals distinct behavioral organization that is missed by traditional assessment criteria.

To complement our quantitative analysis, we provide video files demonstrating the movement tracks of individual mice with overlaid state predictions ("Resting", "Exploring" and "Navigating"). These videos, available as Supplementary Materials, provide direct visualization of the behavioral differences between experimental groups: WT mice showing structured state transitions near the cliff, RD mice displaying uniform exploration patterns, and OF mice exhibiting cliff-independent behavior. The state predictions, derived from our HMM analysis and represented as color-coded trajectories, clearly illustrate how visual depth perception shapes exploratory behavior in real-time.

Having validated our model's ability to detect visually-guided behaviors, we next investigated how varying the properties of visual stimuli affects behavioral responses. We systematically examined two key parameters: the size of the checkerboard pattern and its contrast level. This analysis allowed us to characterize how specific visual features modulate behavioral state transitions and spatial preferences.

## Comparing checkerboard pattern size

We analyzed how mice respond to different sizes of checkerboard patterns (six pattern sizes ranging from 0.1 cm to 8 cm) in the visual cliff setup. The basic movement repertoire remained consistent across all pattern sizes (S5A Fig), with mice exhibiting three distinct behavioral states: "Resting" (~0.4 cm, $CI_{89\%}$: 0.2–0.4), "Exploring" (~1.2 cm, $CI_{89\%}$: 0.7–1.3), and "Navigating" (~2.3 cm, $CI_{89\%}$: 1.7–2.6). However, the pattern size significantly influenced how these states were organized in space, particularly in relation to the cliff boundary (S5B Fig). A key finding emerged in how pattern size affected spatial behavior. While edge influences remained remarkably stable across all conditions (threshold ~3 cm, $CI_{89\%}$: 2.5–3.9), the influence of both center and cliff varied systematically with pattern size (S5C Fig). Smaller patterns (0.1–2 cm) produced more localized responses, with center influence diminishing sharply (~1.5 cm, $CI_{89\%}$: 1.2–1.8 from center). In contrast, larger patterns (6–8 cm) led to more diffuse behavioral responses, with center influence extending over broader regions.

The state change was measured in the same way as in Fig 4. For the horizontal path, regardless of the size pattern, "Navigating" decreased and "Exploring" increased near the center (Fig 5A). We also found that the proportion of "Resting" was highest at the edges. In the difference between two key points ("a" and "b") on the simulated path, "Exploring" was on the increase while "Navigating" was on the decrease in specially larger patterns (6–8 cm).

For the circular path, "Resting" was more prevalent and "Navigating" was less prevalent in all positions in every size (Fig 5B). In the difference between two key points ("a" and "b"), the results varied depending on the size. There was no pattern in the increase or decrease of each state, but behavioral changes were observed at every size. This shows that the cliff effect can be output regardless of the pattern. No size-related characteristics were found in changes in behavioral patterns near the cliff (Fig 5C).

From the above, larger patterns (6–8 cm) elicited stronger cliff effects, characterized by more pronounced transitions between behavioral states near the cliff boundary. This pattern-size dependence suggests that mouse depth perception may be optimally tuned to specific spatial frequencies, with larger patterns potentially providing more salient depth cues.

## Comparing checkerboard pattern contrast

We also analyzed how mice respond to different contrasts (size:2 cm) of checkerboard patterns. Using WT mice, eight contrast conditions varying in total lightness/darkness (the total amount of black) and contrast were analyzed: 1x0.25 (high contrast), 1x0 (maximum contrast), 0x0.75 (high contrast with dark emphasis), 1x0.75 (low contrast with light emphasis), 0x0.25 (low contrast with dark emphasis), and 1x1 (no contrast, completely white).

Interestingly, in contrast with previous analyses where step lengths resolved into three similar states, we observed a gradual increase in step length with total black value (S5D Fig). In lighter conditions (1x1, 1x0.75, 1x0.5, 1x0.25, 1x0, 0x0.75), step lengths were estimated to be about 0.3 ($CI_{89\%}$: 0.2–0.4), 1.2 ($CI_{89\%}$: 0.8–1.4), and 2.3 ($CI_{89\%}$: 1.7–2.7) cm for states 1, 2, and 3 respectively. These increased to 0.4 ($CI_{89\%}$: 0.4–0.5), 1.4 ($CI_{89\%}$: 1.3–1.6), and 2.6 ($CI_{89\%}$: 2.3–2.9) cm for 0x0.5 and 0.4 ($CI_{89\%}$: 0.4–0.5), 1.5 ($CI_{89\%}$: 1.3–1.6), and 2.6 ($CI_{89\%}$: 2.3–2.9) cm for 0x0.25 respectively. The angular parameters ($\rho\_cliff$, $\rho\_edge$, $\rho\_center$), however, mirrored previous findings with no clear changes between conditions (S5E Fig).

S5F Fig illustrates the spatial modulation of edge and center influences across contrast conditions. The edge influence remains robust and consistent across all contrast levels, with a sharp threshold at ~3 ($CI_{89\%}$: 2.7–3.4) cm from the edge. The center influence, however, shows variability across conditions, with highest contrast conditions (1x0, 1x0.25) showing the largest central influence (thresholds at about 12.2 cm and 9.7 cm), whereas other conditions show a threshold ranging from about 3.3 cm to 9.4 cm, but without a clear trend between high and low contrast or lightness and darkness.

The stable state distributions along simulated paths provide insights into how contrast modulates behavioral transitions. For the horizontal path, "Navigating" tends to decrease and "Exploring" tends to increase (Fig 6A). In the difference between the two points (a, b), regardless of the difference in contrast, "Navigating" decreased and "Exploring" increased near the center. On the other hand, for the circular path, "Navigating" decreased and "Resting" increased near the cliff (Fig 6B). This tendency also

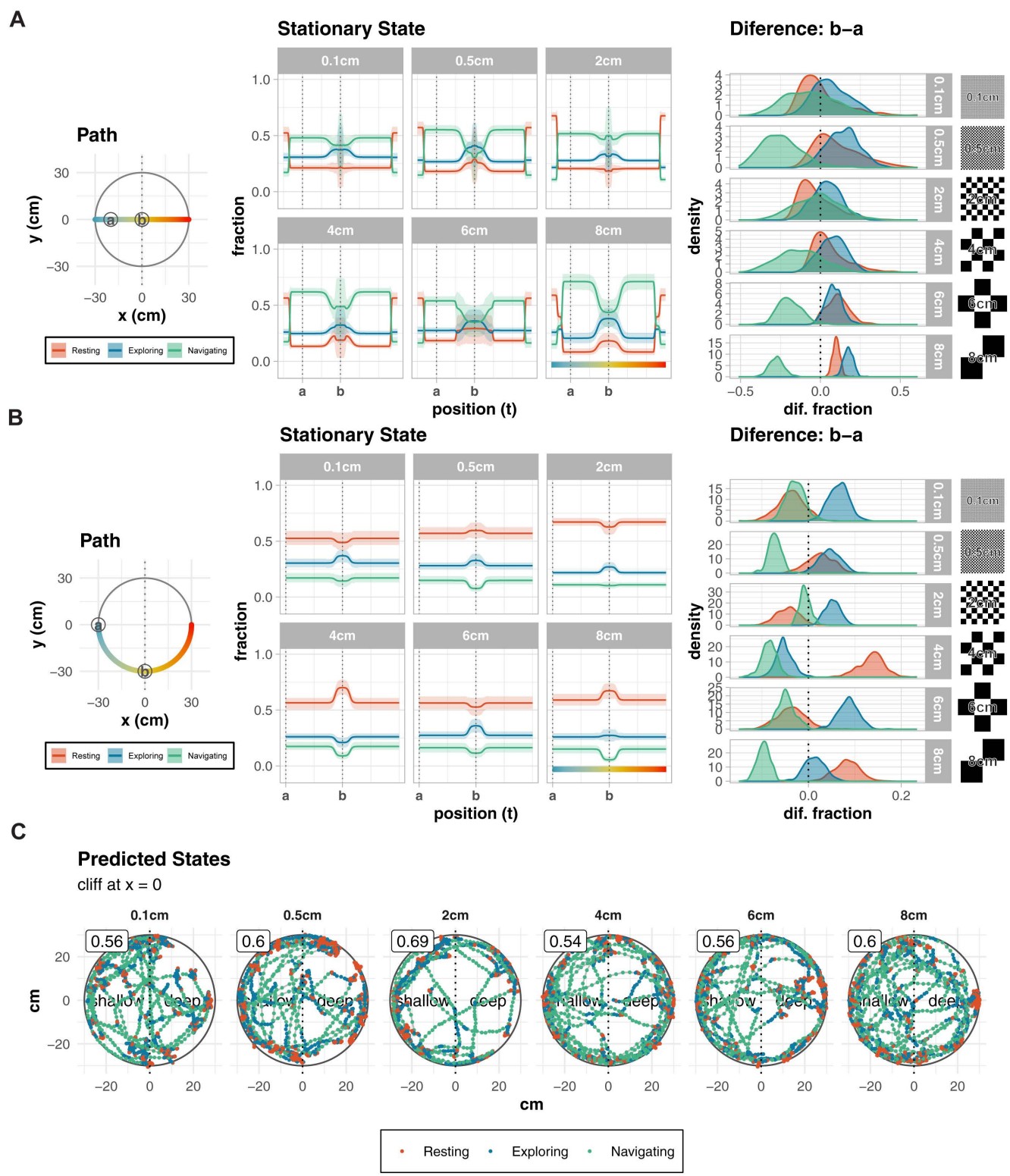

**Fig 5. Behavioral analysis of WT mice under varying checkered pattern sizes in the visual cliff apparatus. (A-B)** Predicted stationary state distributions for simulated movements along two paths: horizontal (A) and a circular path **(B)**. The circular paths explore the interplay between edge and cliff influences, with one path along the edge of the apparatus (radius = 30 cm). (Middle) Stationary state distributions show fractions of states 1 ("Resting",

red), 2 ("Exploring", blue), and 3 ("Navigating", green) as a function of position along the paths. Along the edge (30 cm radius), state 1 dominates due to strong edge influence. (Right) Posterior distributions of state differences between points "a" and "b" highlight the magnitude and nature of transitions induced by the cliff. **(C)** Representative movement trajectories with predicted behavioral states. Points indicate animal position colored by predicted state (red: Resting, blue: Exploring, green: Navigating). Numbers in boxes show shallow side preference rates. Dotted vertical lines mark the cliff boundary (x = 0), dividing shallow and deep sides. There were no differences in characteristics regardless of pattern size.

did not differ depending on the contrast. Surprisingly, higher contrast conditions do not necessarily correlate with stronger cliff effects. As with size, the behavioral patterns did not show any contrast-related characteristics near the cliff (Fig 6C).

### Temporal dynamics of behavior in the visual cliff setup

Understanding how mouse behavior evolves over time is crucial for properly interpreting visual cliff responses. Our temporal analysis across five consecutive intervals (0–3, 3–6, 6–9, 9–12, and 12–15 min) revealed distinct phases of visual cliff response, suggesting a complex interplay between initial depth perception, exploration, and behavioral adaptation.

The most striking temporal pattern emerged in the initial response to the visual cliff (S5G Fig). During the first three minutes, mice exhibited their strongest and most organized behavioral responses, characterized by clear state transitions near the cliff boundary. Step lengths during this period were largest, with "Exploring" showing movements of about 1.2 cm ($CI_{89\%}$: 1.1–1.3) and "Navigating" reaching 2.3 cm ($CI_{89\%}$: 2.2–2.5). The angular concentration parameter ($\rho\_cliff$) for "Exploring" was highest during this period (0.8, $CI_{89\%}$: 0.8–0.9), indicating more directed and purposeful movement patterns (S5H Fig). However, this initial response pattern underwent significant changes over time. Step lengths progressively decreased across all states, with "Exploring" showing the most dramatic reduction from 1.2 cm to about 0.5 cm ($CI_{89\%}$: 0.5–0.6) by the 12–15 min interval. "Navigating" similarly decreased from 2.3 cm to 1.9 cm ($CI_{89\%}$: 1.8–2.1).

The interaction with environmental features also showed distinct temporal patterns (S5I Fig). Edge influence, initially stable at about 3 cm ($CI_{89\%}$: 2.5–3.6), gradually expanded to about 4.5 cm ($CI_{89\%}$: 4.0–5.1) by the final interval. Center influence showed more complex dynamics, remaining localized (1.7–3 cm) during most intervals but unexpectedly expanding to approximately 9.6 cm during the 6–9 min interval. These changes suggest that mice progressively alter their spatial strategy over the course of exploration.

Feature influence profiles revealed distinct temporal patterns. Edge influence, which remained constant at about 3 ($CI_{89\%}$: 2.5–3.6) cm in early intervals, gradually increased to about 4.5 ($CI_{89\%}$: 4.0–5.1) cm by 12–15 min. In contrast, center influence varied more erratically, with localized effects (1.7–3 cm) dominating most intervals but expanding significantly (~9.6 cm) during the 6–9 min interval. These shifting feature influences suggest time-dependent behavioral adjustments to spatial cues.

The stationary state distributions along simulated paths highlighted the interplay between time, feature influences, and state definitions (Fig 7A-7C). Circular paths, which isolate the cliff effect, showed a clear modulation of behavior near the cliff during the 0–3 min interval (Fig 7B). Significant transitions between "Resting" and "Exploring" were observed near the cliff, indicating a strong cliff effect. However, this effect diminished by the 3–6 min interval, as indicated by a lack of state transitions along the same paths. Notably, the cliff effect reemerged gradually from 6−9 min interval to 12–15 min interval, though interpretation is complicated by the progressive reduction in step length and angular concentration parameters. For example, state 2 at 12–15 min, with a mean step length of 0.5 (: 0.5–0.6) cm, is more similar to "Resting" at 0–3 min than to "Exploring" at earlier intervals.

These results demonstrate a dynamic temporal modulation of behavior in the visual cliff setup. While a strong cliff effect is apparent during the initial 0–3 min interval, it diminishes quickly and reappears later (12–15 min). However, progressive changes in state parameters complicate direct comparisons between intervals, as state definitions shift over time. This analysis highlights the importance of considering temporal dynamics and evolving state definitions when interpreting behavioral responses in the visual cliff paradigm.

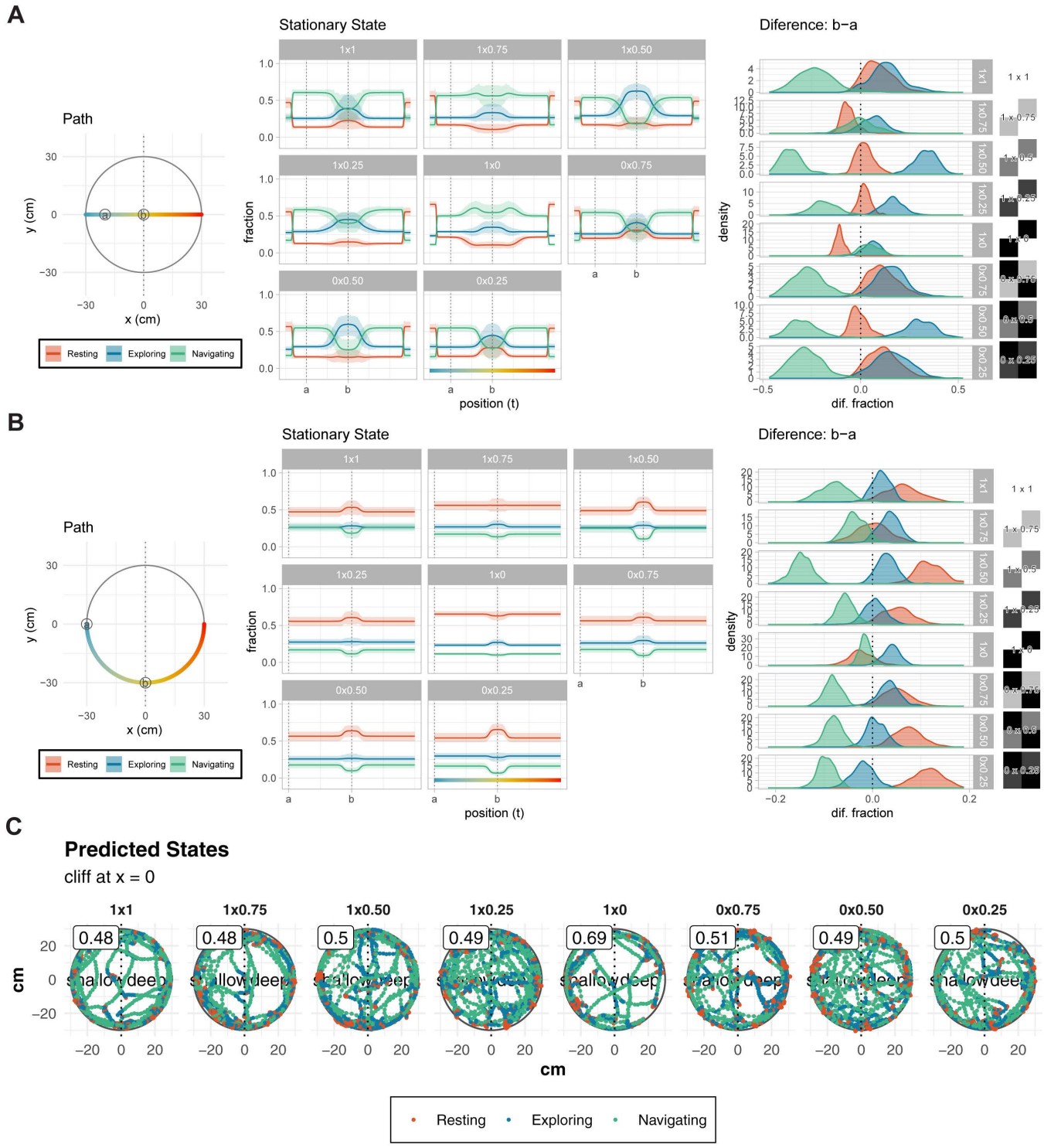

**Fig 6. Behavioral analysis of WT mice under varying contrast conditions in the visual cliff apparatus. (A-B)** Predicted stationary state distributions along simulated movement paths: horizontal (A) and circular **(B)** (radius = 30 cm). Plots depict the fractions of states 1 ("Resting", red), 2 ("Exploring", blue), and 3 ("Navigating", green) as a function of position along the path. Differences between states at key points "a" and "b" along the paths are displayed on the right as posterior distributions. These distributions illustrate significant differences in state transitions, particularly for high-contrast conditions, indicating stronger cliff effects. **(C)** Representative movement trajectories with predicted behavioral states.

Points indicate animal position colored by predicted state (red: Resting, blue: Exploring, green: Navigating). Numbers in boxes show shallow side preference rates. Dotted vertical lines mark the cliff boundary (x = 0), dividing shallow and deep sides. There were no differences in characteristics regardless of pattern contrast.

## Discussion

### Limitations of the shallow side rate and advantages of the HMM framework

Our analysis reveals that mouse responses to visual depth cues are more complex and nuanced than previously recognized through traditional metrics. While the shallow side rate (preference) has been the standard measure in visual cliff studies, our findings demonstrate that identical preference values can arise from fundamentally different behavioral patterns, masking important aspects of visual processing behavior.

The limitations of the shallow side rate become particularly evident when examining individual variations across experimental groups (Fig 8B). In WT mice, shallow side preferences show remarkably high variability, ranging from 0.4 to 0.7. This wide spread could be misinterpreted as suggesting that some individuals have strong cliff avoidance while others prefer the deep side—an interpretation that seems biologically implausible for genetically similar animals. Moreover, while OF controls show expected values around 0.5 (no preference), RD mice paradoxically appear to show negative preferences. These contradictory results highlight how this oversimplified metric can lead to misleading interpretations of visual processing capability.

Our HMM framework resolves these inconsistencies by revealing that visual depth perception manifests through subtle modulations of exploratory behavior rather than simple side preferences. The state-specific cliff effects show tighter distributions within each experimental group, particularly for states 2 ("Exploring") and 3 ("Navigating"), suggesting more reliable detection of true biological responses. This improved consistency is especially important when evaluating subtle behavioral differences or comparing across experimental conditions.

Perhaps most significantly, our approach reveals that mice integrate visual depth information with other spatial cues in ways that cannot be captured by simple preference metrics. This is particularly evident in our pattern size and contrast analyses (Fig 8C-8D). While the traditional metric shows minimal variation across different pattern sizes, our HMM analysis reveals systematic changes in behavioral states, particularly in state 2 ("Exploring"), suggesting more nuanced responses to visual stimuli than previously recognized. These findings indicate that mice process visual depth information through complex behavioral strategies that involve the integration of multiple spatial cues and behavioral states.

The framework's ability to capture temporal dynamics proves particularly valuable for understanding how mice process and adapt to visual depth cues. While the traditional metric shows only modest changes across time intervals, our HMM analysis reveals distinct temporal evolution of behavioral states, particularly during the initial exploration period (0–3 min) compared to later intervals (Fig 8E). This temporal resolution has uncovered previously unrecognized patterns in how mice initially assess and subsequently adapt to visual depth cues.

The HMM framework's key strength lies in its ability to integrate multiple dimensions of behavior while maintaining statistical rigor. By explicitly modeling environmental influences—such as proximity to the cliff, edge, or center—and incorporating temporal dynamics, we capture a more complete picture of how mice process and respond to visual information. This integration allows us to distinguish between seemingly similar behaviors that have different underlying causes, providing new insights into the nature of visual processing in mice.

### Limitations of HMM framework

While our HMM approach addresses many limitations of traditional metrics, it also introduces specific challenges that affect the interpretation of visual processing behavior in mice. These limitations should be carefully considered when drawing biological conclusions.

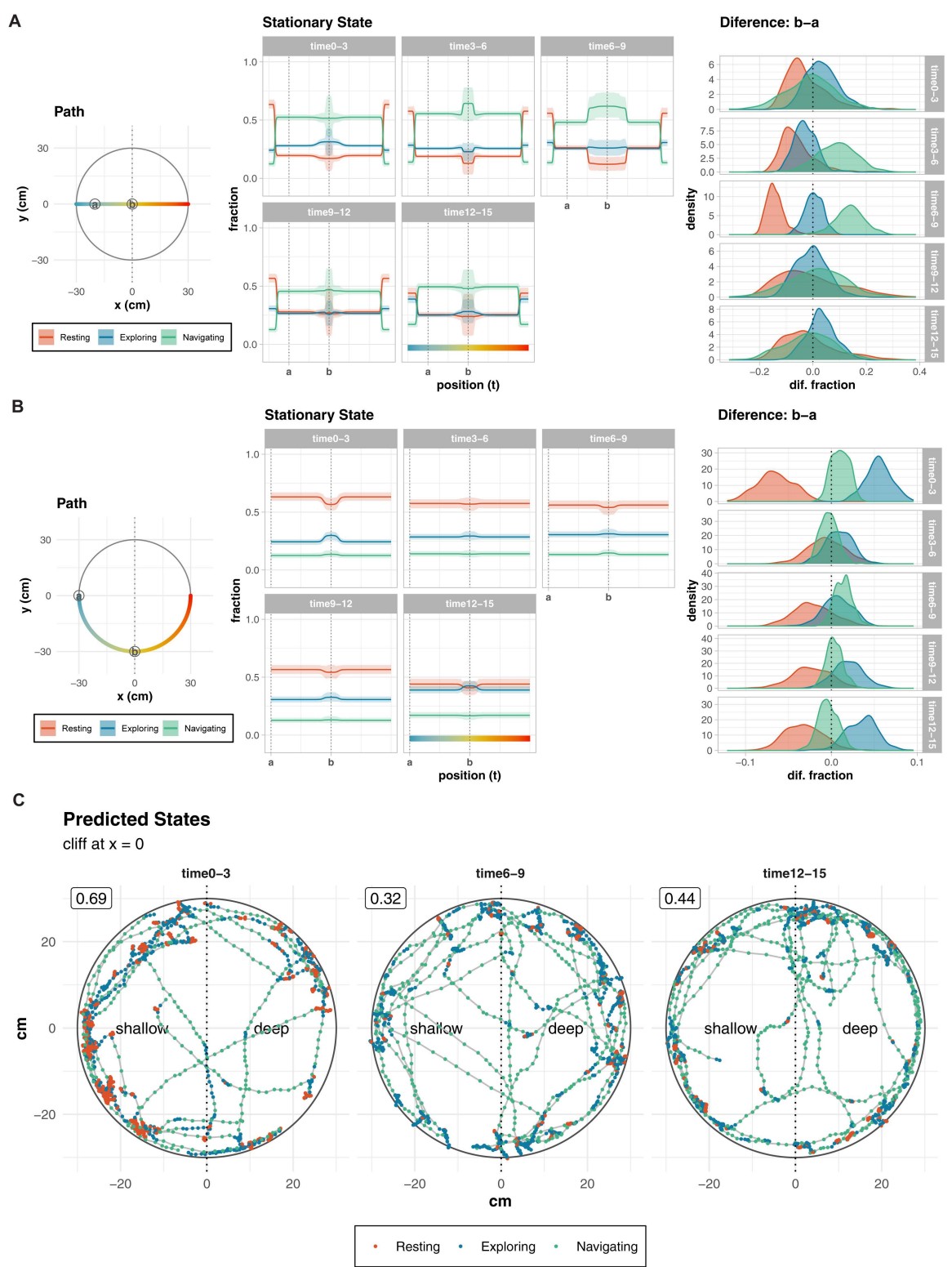

**Fig 7. Temporal analysis of behavioral transitions across five time intervals in the visual cliff setup. (A-B)** Predicted stationary state distributions. Simulated movements along horizontal and circular paths highlight how feature influences and time-dependent state parameters affect stationary state distributions. Circular paths show consistent cliff effects during the 0–3 min interval, with clear transitions in state fractions between points "a" and "b".

However, the cliff effect diminishes during the 6–9 min interval and reappears in the 12–15 min interval, though its interpretation is complicated by changing state definitions (e.g., progressively smaller step lengths). **(C)** Representative movement trajectories with predicted behavioral states. Points indicate animal position colored by predicted state (red: Resting, blue: Exploring, green: Navigating). Numbers in boxes show shallow side preference rates. Dotted vertical lines mark the cliff boundary ($x = 0$), dividing shallow and deep sides. Immediately after the start of the experiment (time 0-3), the proportion of "Resting" participants was high, but as time passed, the proportion decreased, and at times 12-15, "Navigating" participants were more prevalent.

It is important to recognize that the three-state categorization (resting, exploring, navigating) produced by our HMM reflects the specific structure and assumptions of the model. Alternative analytical methods might yield different state definitions or numbers of states. Therefore, the states identified here should be viewed as one of several possible segmentations of locomotor behavior, contingent on the chosen analytical framework. Future work comparing different models may help refine these definitions and advance our understanding of behavioral state dynamics in complex tasks.

**Center and cliff distinction.** A fundamental challenge in our analysis arises from the difficulty in distinguishing between cliff and center influences on behavior. While edge influences are clearly identifiable due to physical constraints, the overlapping spatial domains of cliff and center effects complicate the interpretation of visually-guided behavior. For example, when a mouse changes its movement pattern near the cliff, this could reflect either direct visual perception of depth or a more general response to its position relative to the center of the apparatus.

Although it is not possible to completely separate the cliff and center effects, to properly evaluate the influence of the cliff as much as possible, we fixed the cliff influence parameters ($x_{cliff}$ = 6 cm and $\beta_{cliff}$ = −1) rather than estimating them directly from the data. This decision was driven by our observation that attempting to estimate these parameters led to spurious effects in control conditions (RD and OF groups) where no true cliff response should exist. While this approach successfully prevented false positives in our control groups, it potentially constrains our ability to detect subtle variations in how mice respond to visual depth cues. For instance, if the true spatial range of depth perception extends beyond or falls short of our fixed 6 cm parameter, some visually-guided behaviors might be misattributed to center influence.

**Angular movement modeling.** Our framework successfully captures overall movement patterns but shows systematic deviations in modeling directional behavior, particularly in response to the cliff. Direct observations revealed two distinct types of cliff-avoidance behavior: dramatic trajectory changes away from the cliff and more subtle parallel movements along the cliff boundary. While these behaviors provide compelling visual evidence of depth perception, their relatively rare occurrence made them difficult to capture in our statistical framework.

This limitation reveals an important insight about visual processing: while dramatic cliff avoidance behaviors certainly occur, they may not be the primary way that mice respond to visual depth cues. The effectiveness of our symmetric distance metric, combined with the rarity of overt avoidance behaviors, suggests that subtle modulations of exploratory behavior might be more fundamental to depth perception than dramatic avoidance responses.

These limitations point to broader challenges in behavioral modeling: how to capture important but infrequent behaviors that become statistically diluted in aggregate data, and how to separate overlapping spatial influences on behavior. Future work might address these challenges through additional behavioral covariates, alternative spatial parameterizations, or complementary analytical approaches focused on identifying and characterizing specific behavioral episodes.

## Direct and indirect manifestations of the cliff effect

Our analysis reveals a more nuanced picture of how mice process and respond to visual depth cues than previously recognized. Rather than exhibiting simple avoidance behavior, mice show complex responses that manifest through both direct and indirect pathways, providing new insights into visual processing and spatial behavior.

The cliff effect manifests directly through specific behavioral changes near the cliff boundary. In WT mice, we observe clear transitions between behavioral states as they approach the cliff, typically shifting from "Navigating" to more cautious

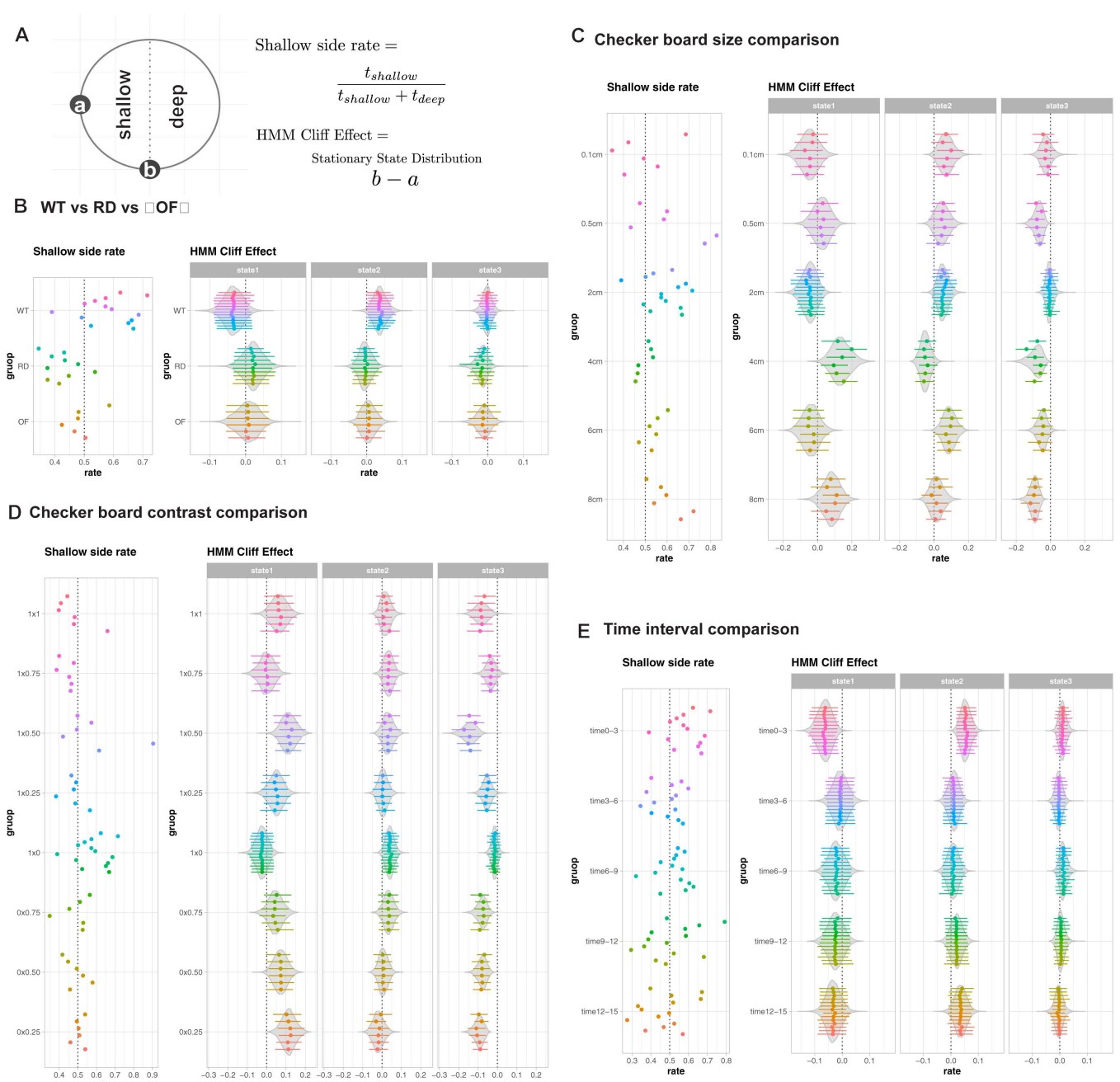

**Fig 8. Comparison of traditional shallow side rate with HMM-derived cliff effect across experimental conditions. (A)** Schematic illustration of measurement methods. Left: The circular apparatus showing points 'a' and 'b' used for stationary state calculations. Right: Equations for shallow side rate (traditional metric) and HMM cliff effect (our novel metric). The shallow side rate is calculated as the proportion of time spent on the shallow side, while the HMM cliff effect represents the difference in stationary state distributions between points 'a' and 'b'. **(B)** Comparison between shallow side rate (left) and HMM cliff effect (right) across three experimental groups: wild-type (WT), retinal degeneration (RD), and open field (OF) conditions. For the shallow side rate, individual data points are shown as dots. For the HMM cliff effect, points and lines represent the mode and 89% credible intervals for individual animals, while violin plots show the group-level posterior distributions. The HMM analysis separates behavioral responses into three distinct states, providing higher resolution of behavioral differences between groups than the traditional metric. **(C)** Effects of checkerboard pattern size (0.1 cm to 8 cm) on behavioral responses. Left: Traditional shallow side rate measurements. Right: HMM cliff effect analysis showing state-specific responses across different pattern sizes. Points and lines represent individual animal estimates (mode and 89% CI), while violin plots show group-level distributions. **(D)** Impact of checkerboard pattern contrast on behavior. Left: Shallow side rate across different contrast conditions (from 0x0.25 to 1x1). Right:

HMM cliff effect analysis showing how contrast modulates different behavioral states. Individual estimates and group-level distributions are shown as in panels B and **C. (E)** Temporal evolution of behavior across five time intervals (0-3 min to 12-15 min). Left: Traditional shallow side preference over time. Right: HMM cliff effect showing how different behavioral states evolve during the experiment. Individual estimates and group-level distributions are shown as in previous panels.

"Exploring" states. These direct responses are validated by their absence in RD mice, confirming their dependence on visual input. However, equally important are indirect effects that emerge through modulation of general exploratory patterns.

A key finding is how visual detection of the cliff reshapes the overall spatial behavior landscape. In WT mice, we observe a sharp decrease in center influence at approximately 6 cm, suggesting that visual detection of the cliff prompts a shift in spatial attention from center exploration to cliff assessment. This interpretation is strengthened by the contrasting behavior of RD mice, whose center influence extends much further (up to 24 cm), reflecting their reliance on tactile rather than visual cues for spatial navigation.

The interaction between direct and indirect effects helps explain seemingly contradictory observations in our data. For example, larger pattern sizes appear to diminish direct cliff responses while simultaneously expanding center influence. Rather than indicating reduced sensitivity to the cliff, this may represent a redistribution of the cliff effect through modified center-based behavior. This interpretation is supported by the intermediate behavior observed in OF mice, where center influence (extending to approximately 12 cm) reflects a balance between visual input and environmental comfort.

These findings highlight a fundamental insight: visual cliff responses should not be conceptualized as isolated behavioral changes but rather as complex redistributions of spatial influences. The apparent trade-off between cliff and center effects demonstrates how mice integrate multiple spatial cues into cohesive behavioral strategies. This integration suggests that depth perception in mice involves sophisticated neural processing that modulates overall exploratory behavior rather than simply triggering avoidance responses.

We acknowledge that while our modeling captures both direct and indirect effects of the cliff, distinguishing between cliff-specific influences and center-based modulations is challenging. This overlap reflects the complex interplay of spatial factors that shape behavioral strategies in mice, and highlights the need for careful interpretation of model outputs when analyzing visually guided locomotion.

## Biological insights from multi-state analysis

Our multi-state analysis has revealed several fundamental aspects of visual processing in mice that extend beyond simple avoidance behaviors. By analyzing how different behavioral states respond to varying visual stimuli over time, we've uncovered new insights into how mice process and utilize visual depth information.

The pattern size experiments provided particularly revealing insights into visual processing mechanisms. While traditional metrics showed minimal systematic variation across pattern sizes, our state-based analysis revealed that different aspects of behavior respond distinctly to pattern size changes. This suggests that depth perception isn't simply stronger or weaker with different pattern sizes, but rather that different components of visual processing—represented by our three behavioral states—are differentially sensitive to spatial frequency information. The observation that larger patterns (6–8 cm) elicit different state transitions compared to smaller patterns (0.1–2 cm) may reflect the involvement of distinct visual processing pathways optimized for different spatial frequencies.

Our contrast analysis revealed another layer of complexity in visual processing. The finding that higher contrast doesn't necessarily correlate with stronger cliff responses challenges simple assumptions about depth perception. Instead, we found that contrast affects the overall organization of exploratory behavior, suggesting that mice integrate contrast information into their broader spatial assessment rather than using it solely for depth perception.

The temporal dynamics analysis revealed perhaps our most surprising finding about visual processing: the evolution of behavioral responses over time suggests a sophisticated learning component to depth perception. Our analysis revealed that the temporal dynamics of behavioral states change over the course of the experiment, consistent with a process of habituation to the environment. Specifically, the initially distinct "Resting", "Exploring" and "Navigating" states began to converge after repeated exposure to the arena, with "Resting" and "Exploring" becoming less distinguishable in later time periods (e.g., 12–15 min; see S5G Fig). This state collapse suggests that mice initially respond strongly to the cliff but gradually adapt their behavior as they become familiar with the environment. Such habituation processes are well documented in behavioral studies and highlight the need to interpret state transitions within the temporal context of the experiment. This also underscores the importance of considering both immediate and time-dependent factors when modeling behavioral dynamics in depth-related tasks.

Comparative analysis of WT and RD mice provided additional insights into the role of vision in spatial behavior. As shown in Fig 4B, the effect size of the cliff perception between WT and RD mice was modest (2–4%). Although this may seem small, it captures behavioral changes that are highly dependent on visual input and demonstrates that vision does indeed impact depth perception. Importantly, even these small effect sizes were accompanied by clear differences in center and edge influences (S4G and S4H Fig). For example, the center influence was substantially different between WT (6.1 cm) and RD (23.6 cm) mice. When considered together, these differences suggest meaningful shifts in spatial behavior that extend beyond simple cliff avoidance. Rather than exhibiting random or undifferentiated movement patterns, RD mice displayed a fundamentally different pattern of spatial exploration, characterized by a broader center influence and reduced edge focus. These findings indicate that visual input not only enables depth perception but also shapes the underlying structure of exploratory behavior. This comparative analysis reveals that even modest effect sizes can reflect biologically significant differences in behavioral strategies, underscoring the complex and multifaceted role of vision in shaping spatial behavior.

These findings have broader implications for our understanding of visual processing in the mammalian brain. The complex interaction between visual input and behavioral state transitions suggests that depth perception involves distributed neural circuits that modulate general behavioral patterns rather than simply triggering reflexive responses. This interpretation aligns with emerging views of visual processing as a distributed, state-dependent process that integrates multiple sources of sensory information to guide behavior.

## Broader implications for behavioral analysis

These biological insights were only possible because of our novel analytical approach, highlighting how methodological advances can drive new biological understanding. While Hidden Markov Models have been successfully employed in ecological studies to analyze movement patterns of tagged animals in the wild, their application to controlled laboratory experiments offers new opportunities for understanding neural mechanisms of behavior.

Our approach of combining carefully designed experimental apparatus with sophisticated analytical tools offers a template for reimagining classical behavioral assays in neuroscience. The elimination of corner effects through circular design demonstrates how thoughtful modification of classical apparatus can significantly improve data quality without compromising the fundamental nature of the test. This principle could be applied to other behavioral paradigms where spatial biases complicate interpretation of results.

The success of our continuous spatial modeling approach, rather than traditional zone-based analysis, has particular relevance for behavioral testing. Many current analyses artificially discretize space into zones of interest, potentially discarding important information about how animals interact with their environment. Our framework demonstrates how continuous spatial modeling can capture subtle behavioral effects while controlling for confounding factors.

Most importantly, our work demonstrates the value of developing generative models of behavior that can separate different influences on animal movement. The combination of precise behavioral tracking and controlled experimental

conditions allows us to test specific hypotheses about how different factors influence behavior, rather than merely describing observed patterns. This represents a significant advance over traditional analyses, bridging the gap between ecological approaches to movement analysis and controlled laboratory experiments.

### Future directions

While our study demonstrates the power of combining refined experimental design with sophisticated analytical methods, it also points to several promising directions for future research.

From a technical perspective, our HMM framework could be enhanced in several ways. The current limitations in modeling angular distributions and distinguishing overlapping spatial influences suggest that more sophisticated movement models might be valuable. Future work could explore additional movement parameters or alternative ways to encode directional biases, particularly for capturing rare but important behaviors like overt cliff avoidance.

Our approach could also be extended to more complex experimental paradigms. While we focused on individual animals in the visual cliff test, the framework could be adapted to analyze how different visual stimuli affect spatial behavior. For example, investigating how variations in lighting conditions, visual patterns, or dynamic stimuli modulate behavioral states could provide deeper insights into visual processing mechanisms.

Finally, integration with neural recording techniques could help link behavioral states to underlying neural activity patterns. Understanding how specific neural circuits give rise to the behavioral states we've identified could provide new insights into the neural basis of visual processing and depth perception.

## Supporting information

**S1 Text.  Hidden Markov Model for Visual Cliff (Supplement).**
(DOCX)

**S1 Fig.  Effects of data thinning on trajectory characteristics. (Top panels)** Trajectory Plots: Example trajectories of a mouse subjected to different thinning levels. **Thin 1** represents unthinned raw data at 30 fps, while **Thin 3** and **Thin 5** show trajectories sampled at effective frame rates of 10 fps and 6 fps, respectively. As thinning increases, finer details such as small, tight turns are omitted, resulting in smoother trajectories. **(Middle panels)** Step Length Distributions: Distributions of step lengths for each thinning level. The horizontal axis is adjusted to account for the increased time intervals due to thinning. While step lengths naturally increase with greater thinning, the overall shape of the distributions remains consistent, indicating that general movement dynamics are preserved. **(Bottom panels)** Angular Difference Distributions: Distributions of angular differences between consecutive steps for each thinning level. These distributions broaden with greater thinning, reflecting larger angular deviations due to the omission of finer trajectory details. This effect corresponds with the smoother paths observed at higher thinning levels.
(PDF)

**S2 Fig.  Mixture model analysis of angular and step length data.** This figure presents the preliminary mixture model analysis conducted to determine the complexity required to represent angular and step length data from mouse trajectories. **(A)** Angular Data Fit: A histogram of observed angular differences (green) is overlaid with fitted two-component mixture models using wrapped Cauchy and von Mises distributions (purple lines). Both distributions capture the sharp central peak (minor directional adjustments) and the broad tails (larger directional shifts) present in the data. Single-component models were insufficient, necessitating at least two components to account for these distinct angular behaviors. **(B)** Angular Components: Estimated distributions of the two mixture components for angular data. The first component (red broader distribution) reflects exploratory movements, while the second (blue narrow peak near zero) represents minor directional adjustments. **(C)** Angular Mixing Ratios: Density plots (shaded areas) and points (individual-level estimates) illustrate the population-level mixture proportions ($\theta_0$) for the two angular models. The wrapped Cauchy model allocates

approximately 20% to the broad component, whereas the von Mises model assigns ~40%, reflecting differences in how each model handles heavy tails. Although predictive performance differences were minor, the wrapped Cauchy model's slight advantage informed our choice for HMM modeling. **(D)** Step Length Data Fit: A histogram of observed step lengths (green) is shown with fitted two- and three-component mixture models using Gamma and Lognormal distributions (purple lines). The step length data exhibit multiple peaks, and while two-component models improved upon single-component fits, a three-component Gamma model was necessary to fully capture the multimodal nature of movement intensities. **(E)** Step Length Components: The three-component Gamma model identifies distinct states of movement intensity: short step lengths (red, minimal movement), intermediate step lengths (blue, exploratory movement), and long step lengths (green, active locomotion). The Lognormal models show less distinct separation among components. **(F)** Step Length Mixing Ratios: Density plots and points display the population-level mixture proportions for the two- and three-component Gamma models, with the three-component version producing more balanced and interpretable mixtures. Although these mixture components do not directly define hidden behavioral states, they guided the decision to use multiple states and appropriate distributions in the subsequent Hidden Markov Model (HMM) analysis.
(PDF)

**S3 Fig. Mouse tracks and predicted states in the visual cliff experiment. (top)** Representative tracks of mice from three experimental groups: wild-type (WT), retinal degeneration model (RD), and wild-type in an open field setup (OF, no visual cliff). Tracks are color-coded to indicate continuous recordings, with each track representing the movement of an individual mouse over time. The tracks illustrate the spatial behavior of each group, with cliff locations indicated along the x-axis (cliff at $x = 0 x = 0 x = 0$). **(bottom)** Predicted behavioral states superimposed on the tracks. States are color-coded: "Resting" (red), "Exploring" (blue), and "Navigating" (green). The predicted states reveal the spatial distribution of behavioral patterns across the apparatus and highlight differences between experimental groups. WT tracks show a higher proportion of "Navigating" states near the cliff edge, indicating active exploration and interaction with the cliff. In contrast, RD mice display more "Resting" states throughout the apparatus, reflecting diminished visual input and exploratory behavior. OF mice exhibit a combination of "Exploring" and "Navigating" states, with reduced cliff-specific effects due to the absence of depth cues.
(PDF)

**S4 Fig. Posterior parameter estimates for the Hidden Markov Model across WT, RD, and OF groups. (A, B)** Step length parameters: Posterior distributions of the step length parameters for the three behavioral states ("Resting", "Exploring" and "Navigating") are shown for wild-type (WT), retinal degeneration (RD), and open field (OF) groups. WT mice exhibit distinct separation between states, while RD and OF groups show overlapping distributions, particularly for states 1 and 2, reflecting less structured movement patterns. **(C~F)** Angular concentration parameters: These parameters describe the directional persistence for each state under basal conditions and near environmental features (cliff, edge, center). WT mice display sharper directional persistence compared to RD and OF groups, particularly near the cliff, highlighting their reliance on visual depth cues. **(G~J)** Feature-specific influence parameters: Posterior distributions of the slope and offset parameters for center and edge influences are shown. WT mice demonstrate smaller center offsets compared to RD and OF groups, reflecting a stronger and more localized center influence. Edge parameters remain consistent across groups, underscoring the uniform effect of physical barriers. **(K, L)** Transition probability matrix (TPM) regression coefficients: Group-level and individual-specific posterior distributions of TPM regression coefficients are shown, highlighting the influence of spatial covariates (cliff, edge, center) on transitions between behavioral states. WT mice exhibit stronger spatial modulation in transitions, particularly between states 1 ("Resting") and 2 ("Exploring"), while RD and OF mice show flatter distributions, indicating weaker transitions driven by spatial cues. This figure summarizes the posterior distributions of key HMM parameters, providing insights into how spatial features and behavioral states are modulated across WT, RD, and OF groups. The differences in posterior estimates underscore the distinct behavioral strategies employed by each group in response to the visual cliff apparatus.
(PDF)

**S5 Fig. Summary of predicted distribution of step length and angle, and feature influence profiles for edge and center in some conditions. (A)** Predicted distributions of step lengths for the three behavioral states ("Resting", red; "Exploring", blue; "Navigating", green) across all pattern sizes. Solid lines represent the mean predicted distributions, while shaded regions indicate the 89% confidence intervals. Step lengths for the three states remain consistent across pattern sizes, with a stationary state ("Resting") at approximately 0.1 cm, an intermediate exploratory state ("Exploring") at ~1.1 cm, and a high-movement state ("Navigating") at ~2.3 cm. **(B)** Predicted angular distributions for the three behavioral states under varying checkered pattern sizes. The angular range is restricted to $[-\pi/8, \pi/8]$ to highlight differences in concentration across conditions. A dashed line indicates the uniform angle density ($y = 0.159$) for reference. **(C)** Feature influence profiles for edge (left) and center (right) across pattern sizes. Edge influence remains stable across all conditions, with a consistent threshold at ~3 cm from the edge. Center influence, however, expands significantly with increasing pattern size. For smaller patterns (e.g., 2 cm), center influence diminishes ~4 cm from the center, whereas for larger patterns (6 cm and 8 cm), the influence extends outward, reflecting broader modulation of behavior. **(D-E)** Predicted distributions of step lengths (B) and angular differences (C) for each contrast condition. For step lengths, distributions of the three behavioral states ("Resting", red; "Exploring", blue; "Navigating", green) are shown with lines representing the mean and shaded areas representing the 89% confidence interval. The angular distributions, restricted to the range $[-\pi/8, \pi/8]$, illustrate the concentration of angular differences for basal conditions and near spatial features (cliff, edge, center). A dotted line indicates a uniform angular distribution ($y = 0.159$). **(F)** Feature influence profiles for edge and center across contrast conditions. Edge influence (left) remains stable and robust, with a consistent threshold of ~3 cm across all conditions. Center influence (right) varies across contrast conditions, with no systematic trend between high and low contrast levels. **(G-H)** Predicted step length (G) and angular (H) distributions for time transition. Step lengths were resolved into three discrete states but the mean step lengths for each state decreased over time, indicating a progressive reduction in movement intensity across the intervals (0–3 min, 3–6 min, 6–9 min, 9–12 min, and 12–15 min). Shaded areas indicate 89% confidence intervals. Angular distributions for the basal condition and near environmental features (cliff, edge, center) are shown for each time interval. The dotted line represents a uniform angular distribution. While angular concentration parameters (ρ) remained stable for some states (e.g., State 3 ("Navigating"), State 2 ("Exploring") exhibited a noticeable decline in concentration values over time, particularly for the cliff condition. **(I)** Feature influence profiles. Edge influence (left) increased gradually over time, with the threshold distance expanding from ~3 cm to ~4.5 cm. In contrast, center influence (right) was more variable, remaining localized (1.7–3 cm) in most intervals but showing a broader range (~9.6 cm) at 6–9 min.
(PDF)

**S1 Movie. Top-down video of a WT mouse in the standard visual cliff task.** Colored dots show inferred behavioral states: red = Resting, blue = Exploring, green = Navigating.
(MOV)

**S2 Movie. Top-down video of a WT mouse in the open field (no visual cliff).** Colored dots show inferred behavioral states: red = Resting, blue = Exploring, green = Navigating.
(MOV)

**S3 Movie. Top-down video of a RD mouse in the visual cliff task.** Colored dots show inferred behavioral states: red = Resting, blue = Exploring, green = Navigating.
(MOV)

## Author contributions

**Conceptualization:** Hironobu Shuto, Take Matsuyama.

**Data curation:** Hironobu Shuto, Toshiki Maeda, Take Matsuyama.

**Formal analysis:** Hironobu Shuto, Take Matsuyama.

**Funding acquisition:** Take Matsuyama.

**Investigation:** Hironobu Shuto, Toshiki Maeda, Take Matsuyama.

**Methodology:** Hironobu Shuto, Take Matsuyama.

**Project administration:** Take Matsuyama.

**Supervision:** Chieko Koike, Masayo Takahashi, Michiko Mandai, Take Matsuyama.

**Validation:** Hironobu Shuto.

**Visualization:** Hironobu Shuto.

**Writing – original draft:** Hironobu Shuto, Take Matsuyama.

**Writing – review & editing:** Hironobu Shuto, Take Matsuyama.

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
