## [Decision Letter · Decision Letter 0]

27 May 2025

Dear Dr. Matsuyama,

Thank you for submitting your manuscript to PLOS ONE. After careful consideration, we feel that it has merit but does not fully meet PLOS ONE’s publication criteria as it currently stands. Therefore, we invite you to submit a revised version of the manuscript that addresses the points raised during the review process.

We look forward to receiving your revised manuscript.

Kind regards,

Mario Treviño Villegas, Ph.D

Academic Editor

PLOS ONE

Journal Requirements:

[This work was supported by JSPS KAKENHI Grant Number 24H00747 awarded to TM].

5. Please respond by return e-mail so that we can expand the acronym “JSPS” in your financial disclosure so that it states the name of your funders in full. 

We will amend your financial disclosure and competing interests on your behalf.

6. Thank you for stating the following in the Competing Interests section:

[Authors with competing interests

TM, SH, MT are employees of Vision Care Inc., a startup working on the development of treatments for vision restoration.].

Reviewers' comments:

Reviewer's Responses to Questions

**Comments to the Author**

1. Is the manuscript technically sound, and do the data support the conclusions?

Reviewer #1: Partly

Reviewer #2: Yes

Reviewer #3: Yes

2. Has the statistical analysis been performed appropriately and rigorously?

Reviewer #1: I Don't Know

Reviewer #2: Yes

Reviewer #3: Yes

3. Have the authors made all data underlying the findings in their manuscript fully available?

Reviewer #1: Yes

Reviewer #2: Yes

Reviewer #3: Yes

4. Is the manuscript presented in an intelligible fashion and written in standard English?

Reviewer #1: Yes

Reviewer #2: Yes

Reviewer #3: Yes

Reviewer #1: By addressing these points comprehensively, the authors can significantly improve the manuscript and increase its potential for publication.

Major comments.

1.Novelty and Originality

Novel research can be best described as one or more elements of research that are unique, such as a new methodology or a new observation that leads to the acquisition of new knowledge. It is this novelty that contributes to scientific progress. Since the main aim of research is to unravel what is unknown or to challenge views or ideas that may or may not be based on sound scientific principles, this exclusivity of novel research allows us to expand our horizons beyond the realms of known domains. It seems that the current study lacks the necessary novelty and originality.

2.Abstract

The abstract serves as the first impression of a manuscript for prospective readers, including the editor and reviewers, and should, therefore, be drafted with great care. Decisions to proceed with the peer review process are often influenced by the clarity of the information presented in the abstract. Readers may include researchers and potential authors who will cite the article, or they may not be researchers but have an interest in the topic; thus, an effective abstract is crucial for attracting a broader readership.

It is essential to be succinct and precise in writing, providing a clear and comprehensible summary of the study. Authors should articulate a clear and concise aim, describe the methodology, including study design, setting, and population, and ensure that the results (key findings) presented in the abstract (and the manuscript) align with this aim. Conclusions and interpretations must be supported by the study findings. Additionally, it is important to avoid jargon, uncommon abbreviations, and references in the abstract.

The abstract of the present article is written in a general manner and requires refinement to enhance its scientific clarity, allowing readers to better connect with the key concepts and findings of the manuscript.

3. Introduction section of the manuscript

The introduction does not effectively establish the rationale for the study and requires strengthening. It lacks clarity in presenting the importance of the research, the existing knowledge on the topic, and the identified gap that the study aims to address. Additionally, it should provide more background information to support the rationale and better articulate the specific research aims.

In the introduction section, the authors of the article should provide a compelling and concise explanation of why the topic is important and useful within the field of animal studies, what is known about the topic, and what research gaps exist. The scientific rationale and innovative aspects of the study should also be clearly articulated. The originality and innovation of the study need to be justified and targeted. Authors should demonstrate awareness of seminal publications in this area and incorporate recent literature and any systematic reviews into their work.

Additionally, authors should present both an international and national perspective and refrain from providing a historical account. The information presented should guide readers toward the aims and objectives, which are included at the end of the introduction. A key point is to avoid exaggerated claims or expressions such as “innovative,” “for the first time,” and “first ever,” unless this is indeed the case.

The introduction section should be approximately three to five paragraphs in length. Look at examples from your target journal to decide the appropriate length. This section begins with a general context, narrowing to the specific focus of the paper.

Include five main elements: why your research is important, what is already known about the topic, the “gap” or what is not yet known about the topic, why it is important to learn the new information that your research adds, and the specific research aim(s) that your paper addresses. Your research aim should address the gap you identified. Be sure to add enough background information to enable readers to understand your study.

Identify the knowledge gap and explain how this knowledge gap will be addressed in this study. Additionally, the literature review and theoretical framework require further strengthening and coherence.

4.Methods section of the manuscript

The methods section should provide a clear, sufficiently detailed, and reproducible description of how the study was conducted to allow for replication.

The methods section lacks sufficient clarity, detail, and transparency to allow for replication. Concerns exist regarding the internal validity of the instruments used.

The methods have not been described clearly and transparently (e.g., data collection, interventions, and statistical analyses). The description of the methods is not sufficiently clear and comprehensive to allow for the reproduction of this study, and there are concerns regarding the internal validity of the instruments used.

5. Discussion section of the manuscript

In the discussion section, authors should address what the results mean and how this work contributes to advancing the field of animal studies in the specified area. Authors should begin with a clear and grounded summary of the key findings to ensure that focus is maintained and the study aims/objectives are addressed. This section should proceed with interpretation, rather than reiteration, of the presented results in the context of published literature. Results should be related to those of similar studies, and efforts should be made to explain why similar or contradictory results were obtained.

Points to consider include avoiding statements that go beyond what the results can support and refraining from the sudden introduction of new terms or ideas. Authors may speculate on possible interpretations; however, these should be rooted in fact. Authors should indicate the study’s strengths without overemphasizing, discuss the implications for practice, and put forward recommendations for future work/research. The strengths and limitations of the study, and how they impact the generalizability/transferability of findings, should be discussed.

A clear, concise, and convincing conclusion that aligns with the study’s objectives and results, and reinforces the significance of the research and its implications for practice, should be presented. The results may not be generalizable/transferable to other study populations, so words such as “may” should be used when extrapolating results.

-Opposite the introduction section, the discussion should take the form of a right-side-up triangle beginning with interpretation of your results and moving to general implications.

-This section typically begins with a restatement of the main findings, which can usually be accomplished with a few carefully-crafted sentences. Next, interpret the meaning or explain the significance of your results, lifting the reader’s gaze from the study’s specific findings to more general applications. Then, compare these study findings with another research. Are these findings in agreement or disagreement with those from other studies? Does this study impart additional nuance to well-accepted theories? Situate your findings within the broader context of scientific literature, then explain the pathways or mechanisms that might give rise to, or explain, the results.

-The next element of the discussion is a summary of the potential impacts and applications of the research. Should these results be used to optimally design an intervention? Does the work have implications for clinical protocols or public policy? These considerations will help the reader to further grasp the possible impacts of the presented work. Finally, the discussion should conclude with specific suggestions for future work. Here, you have an opportunity to illuminate specific gaps in the literature that compel further study. Avoid the phrase “future research is necessary” because the recommendation is too general to be helpful to readers. Instead, provide substantive and specific recommendations for future studies.

Reviewer #2: The article is well-written and well-structured. The authors innovatively combined classic test models to avoid the limitations of using the models individually. Hence, the results of the authors' model can provide new inisght that help us understand the depth perception and related cognitive processes of mice. Good work! Before being considerated to be publihsed, the article still need some minor improvements concentrated on its writing and formality. Please see my comments below:

Abstract: I am not sure about what the maximum keyword limit is. For clarity and to make it easier for readers to pinpoint the focus of the article, please remove some keywords that have overlapping meanings (in the scientific sense).

Line 2-9: The first sentence of the abstract lacks clarity. You suggest that “Understanding how mice process and respond to visual depth cues is crucial for studying visual perception, but traditional behavioral analyses often miss subtle yet important aspects of this process”. What are the important aspects? How are those aspects related to the cues and processes you mentioned in the following sentences? Please clarify.

Line 36-38: You suggest that “Simpler tests such as the optomotor response provide basic measures of visual acuity but fail to capture higher-order cognitive processes.” One should be cautious when using terms like cognitive or higher-order cognitive processes, which may imply intentionality and cross-species similarities.

Line 130-132: You write “Our results demonstrate that this integrated approach provides deeper insights into visual processing than traditional methods, while offering a robust framework for quantitative behavioral analysis”. Using “while” in this context is not appropriate and causes confusion. Consider changing “while” to “and”.

Line 149: Consider adding photos of the apparatus.

Start from line 134: Please note the inconsistencies of parameters (e.g., cm, mm, min). It is not appropriate to use abbreviations from the start without defining them. Instead, I suggest using the complete spelling of the parameters when they first appear, then adding abbreviations in brackets as a definition or explanation.

Line 159: Is “wildtype mice” an appropriate term? Please also include examples of species and their common and Latin names.

Reviewer #3: The manuscript entitled "Hidden Markov Models Reveal Complex Depth Processing in Mouse Visual Behavior" presents a study that analyzes how mice adjust their locomotor behavior in response to visual cues of depth, using a circular open field setup and a probabilistic modeling approach. This is a thought-provoking and methodologically innovative article, with several relevant contributions, and I have genuinely enjoyed reading it.

That said, I believe there are a number of aspects that could be reconsidered or further elaborated in order to deliver a more complete and balanced article for the readers of PLOS ONE. I outline these considerations below.

Conceptual Framework and Title

First, I would like to draw attention to the title of the article. Not because of its wording per se, but because it may point to a conceptual framing that could be refined. The phrase "…reveal complex depth processing" seems to imply a demonstration of internal perceptual or neural processes, yet the study does not directly measure perceptual mechanisms or neural responses. Rather, it investigates how behavior—specifically, the balance between exploratory and defensive responses—is modulated as the animal approaches a visually defined stimulus (the "cliff") in space. The title may therefore overstate the nature of the findings. A more accurate and transparent formulation would be one that reflects the behavioral and modeling focus of the work, such as highlighting state transitions, locomotor patterns, or probabilistic classification in depth-based tasks.

Along these lines, I believe the introduction could be strengthened by explicitly framing the study in terms of the functional interaction between exploratory and defensive tendencies. These behavioral systems are known to operate in tension, especially in ambiguous or risky environments. As the organism nears a potentially threatening stimulus, defensive responses tend to dominate; however, with repeated exposures, a degree of habituation can shift this balance back toward exploration. This dynamic is central to the interpretation of the results, and acknowledging it early on could help position the study within broader frameworks of behavior regulation, attention, and adaptation.

Experimental Design

The circular open field design is a clear strength of the study. It elegantly avoids the corner biases that often confound results in square arenas and encourages more uniform spatial exploration. However, it would be helpful if the authors clarified whether this setup was controlled for potential confounds such as differences in lighting, floor reflections, or subtle asymmetries in visual patterns, which might influence movement patterns near the edges.

Methodological Considerations

Regarding the analysis method, the use of Hidden Markov Models (HMMs) is one of the most distinctive features of the study. Yet the rationale for choosing this specific approach could be articulated more explicitly. Since the identification of behavioral patterns depends to a great extent on the analytical framework employed, it is important for the reader to understand why this model was favored—particularly whether it was selected a priori, or after comparing it to other options. Were alternative unsupervised learning methods (e.g., k-means clustering, PCA-based trajectory analysis, or duration-aware segmentation models) considered and rejected? What specific advantage does the HMM provide in terms of interpretability or fit to the biological data? Addressing these questions would help readers appreciate the methodological decision and interpret the resulting "three-state" model with the appropriate caution.

Relatedly, it is worth noting that the categorization into three discrete states—resting, exploring, navigating—is a product of the model architecture itself. While these categories are intuitive and align well with observed behavior, it would be valuable to acknowledge that other methods might yield different numbers or types of states. This does not diminish the value of the results, but it does suggest that the findings should be interpreted as model-dependent, and potentially open to alternative decompositions of behavior. A brief discussion on this point would demonstrate awareness of the limitations of statistical segmentation and encourage future comparative validation.

Additional Methodological Questions

Additionally, I would like to raise several specific questions that arose during my reading of the manuscript. While some of these may reflect gaps in my own understanding of the methodology, I believe they represent queries that could occur to readers of PLOS ONE, and addressing them explicitly in the text would enhance the manuscript's clarity and reproducibility.

Methodological specifications:

The rationale for tracking 12 body points when only the body center coordinates are used in the analysis could be clarified. Were the additional points considered for future analyses, or do they contribute to tracking accuracy?

The 90% confidence threshold for DeepLabCut tracking appears somewhat arbitrary—was this threshold validated against manual scoring or compared to alternative cut-offs?

While the 3000K lighting choice is explained by analogy to dawn/dusk conditions, it would be valuable to know whether this parameter was empirically validated or if alternative color temperatures were tested.

Experimental design considerations:

The protocol mentions that mice always start facing the shallow side—was counterbalancing of starting orientation considered to control for potential directional biases?

The procedure for "gently guiding" non-responsive mice could introduce variability—how frequently was this intervention required, and were these trials analyzed separately?

Results interpretation and presentation:

The authors acknowledge fixing certain spatial parameters (cliff influence at 6 cm) to prevent spurious effects in control groups. While this approach has merit, I wonder if it might limit the detection of genuine individual or group differences in spatial sensitivity to visual cues.

The discussion mentions that "dramatic cliff avoidance behaviors certainly occur" but are "relatively rare," which seems to contradict some of the stronger claims about depth processing complexity. Could the authors clarify how these infrequent but compelling behaviors relate to their overall model?

The temporal analysis reveals interesting dynamics where the cliff effect diminishes and reappears over time. I wonder if this pattern might reflect habituation processes rather than sustained depth processing—could the authors comment on this interpretation?

The behavioral state definitions appear to shift over time (e.g., state 2 at 12-15 min resembling early "Resting" more than "Exploring"). While the authors acknowledge this complexity, it might be helpful to discuss how this affects the biological interpretation of the states.

The integration of multiple spatial influences (cliff, center, edge) is intriguing, but the acknowledged difficulty in distinguishing between cliff and center effects raises questions about the specificity of the visual depth response.

Some key findings rely on relatively small effect sizes (2-4% differences). While statistically robust, could the authors comment on the potential biological significance of these magnitudes?

Given that many crucial results are presented in supplementary materials, readers might benefit from having some of these key findings highlighted more prominently in the main text.

Summary

In summary, this manuscript presents a compelling and well-executed study that combines a thoughtful behavioral design with a sophisticated modeling approach. The findings are original and relevant, particularly in their attempt to move beyond traditional binary behavioral metrics. I believe that with some refinement in how the framework is presented—both conceptually and methodologically—the article could reach a broader readership and have a greater impact. I hope the authors find these comments constructive in strengthening an already promising piece of work.

**Do you want your identity to be public for this peer review?** For information about this choice, including consent withdrawal, please see our Privacy Policy

Reviewer #1: No

Reviewer #2: No

Reviewer #3: No

---

## [Author Response · Author response to Decision Letter 1]

1 Jul 2025

Response to Reviewers

Responses to Reviewer #2

Reviewer #2: The article is well-written and well-structured. The authors innovatively combined classic test models to avoid the limitations of using the models individually. Hence, the results of the authors' model can provide new inisght that help us understand the depth perception and related cognitive processes of mice. Good work! Before being considerated to be publihsed, the article still need some minor improvements concentrated on its writing and formality. Please see my comments below:

We thank the reviewer for their time, effort, and constructive feedback on our manuscript. We appreciate the positive comments regarding the overall structure, the innovative combination of classic test models, and the relevance of our results for understanding depth perception and related processes in mice. We have carefully considered the reviewer’s suggestions regarding minor improvements in writing and formality, and we have addressed each point below.

Abstract: I am not sure about what the maximum keyword limit is. For clarity and to make it easier for readers to pinpoint the focus of the article, please remove some keywords that have overlapping meanings (in the scientific sense).

We have revised the keywords section by removing overlapping terms to enhance clarity and focus.

Line 2-9: The first sentence of the abstract lacks clarity. You suggest that “Understanding how mice process and respond to visual depth cues is crucial for studying visual perception, but traditional behavioral analyses often miss subtle yet important aspects of this process”. What are the important aspects? How are those aspects related to the cues and processes you mentioned in the following sentences? Please clarify.

We appreciate the comment. We agree that the first sentence could be clearer regarding the important aspects of visual depth processing that traditional analyses might miss. We have revised the first sentence of the abstract to clarify that the important aspects include temporal dynamics, state transitions, and integration of multiple spatial cues—all of which are crucial for understanding how mice process depth information. We also now directly link these aspects to the methods and findings presented in the manuscript. This sentence was replaced with the following:

“Understanding how mice process and respond to visual depth cues is crucial for studying visual perception, yet traditional behavioral analyses often miss key aspects of this process, such as the dynamic transitions between behavioral states, the influence of environmental context, and the integration of multiple spatial cues that shape depth-related behaviors. “

Line 36-38: You suggest that “Simpler tests such as the optomotor response provide basic measures of visual acuity but fail to capture higher-order cognitive processes.” One should be cautious when using terms like cognitive or higher-order cognitive processes, which may imply intentionality and cross-species similarities.

We have revised the sentence to avoid implying intentionality or strong cross-species cognitive parallels, and to better align with a cautious, comparative perspective.

Simpler tests such as the optomotor response provide basic measures of visual acuity but may not fully capture more complex behavioral adaptations and context-dependent responses.

Line 130-132: You write “Our results demonstrate that this integrated approach provides deeper insights into visual processing than traditional methods, while offering a robust framework for quantitative behavioral analysis”. Using “while” in this context is not appropriate and causes confusion. Consider changing “while” to “and”.

We have replaced "while" with "and offer" to improve clarity in this sentence.

Line 149: Consider adding photos of the apparatus.

A photograph of the apparatus is already included in Figure 1A. We have ensured that this figure clearly shows the apparatus as suggested.

Start from line 134: Please note the inconsistencies of parameters (e.g., cm, mm, min). It is not appropriate to use abbreviations from the start without defining them. Instead, I suggest using the complete spelling of the parameters when they first appear, then adding abbreviations in brackets as a definition or explanation.

We revised all measurements to use millimeters (mm) for consistency, and defined all abbreviations at their first occurrence to ensure clarity throughout the manuscript.

Line 159: Is “wildtype mice” an appropriate term? Please also include examples of species and their common and Latin names.

“Wildtype” is a commonly used term in the field, especially when contrasting it with the retinal degeneration model (RD). Nevertheless, we have revised the text to include the proper scientific names for both: C57BL/6J for the WT mice and rd1 for the RD mice.

“Comparisons between wild-type C57BL/6J mice (Mus musculus), retinal degeneration models (rd1-2J, C57BL/6J background, Mus musculus), and control conditions confirm that these behavioral patterns specifically reflect visual processing rather than general exploratory behavior.”

Responses to Reviewer #3

The manuscript entitled "Hidden Markov Models Reveal Complex Depth Processing in Mouse Visual Behavior" presents a study that analyzes how mice adjust their locomotor behavior in response to visual cues of depth, using a circular open field setup and a probabilistic modeling approach. This is a thought-provoking and methodologically innovative article, with several relevant contributions, and I have genuinely enjoyed reading it.

That said, I believe there are a number of aspects that could be reconsidered or further elaborated in order to deliver a more complete and balanced article for the readers of PLOS ONE. I outline these considerations below.

We appreciate the reviewer’s thorough and thoughtful comments, as well as their positive assessment of the methodological innovation and relevance of our study. We have carefully considered each of the reviewer’s suggestions and have made revisions to address the concerns and improve the clarity, rigor, and balance of the manuscript. Our point-by-point responses to the specific comments are provided below.

Conceptual Framework and Title

First, I would like to draw attention to the title of the article. Not because of its wording per se, but because it may point to a conceptual framing that could be refined. The phrase "…reveal complex depth processing" seems to imply a demonstration of internal perceptual or neural processes, yet the study does not directly measure perceptual mechanisms or neural responses. Rather, it investigates how behavior—specifically, the balance between exploratory and defensive responses—is modulated as the animal approaches a visually defined stimulus (the "cliff") in space. The title may therefore overstate the nature of the findings. A more accurate and transparent formulation would be one that reflects the behavioral and modeling focus of the work, such as highlighting state transitions, locomotor patterns, or probabilistic classification in depth-based tasks.

We appreciate this thoughtful comment. To better align the title with the study’s focus on behavioral state transitions and probabilistic modeling, we have revised the title from:

“Hidden Markov Models Reveal Complex Depth Processing in Mouse Visual Behavior”

to

“Hidden Markov Models Reveal Behavioral State Dynamics in Depth-Related Locomotion in Mice.”

We believe this new title more accurately reflects the study’s emphasis on state transitions and behavioral patterns rather than implying direct measurement of internal perceptual processes. Additionally, we have revised the short title from:

“HMM Analysis of Mouse Visual Depth Processing” to “HMM Analysis of Behavioral State Dynamics in Mice”.

We believe these changes align well with the reviewer’s suggestion to clarify the focus and avoid overstating the study’s claims.

Along these lines, I believe the introduction could be strengthened by explicitly framing the study in terms of the functional interaction between exploratory and defensive tendencies. These behavioral systems are known to operate in tension, especially in ambiguous or risky environments. As the organism nears a potentially threatening stimulus, defensive responses tend to dominate; however, with repeated exposures, a degree of habituation can shift this balance back toward exploration. This dynamic is central to the interpretation of the results, and acknowledging it early on could help position the study within broader frameworks of behavior regulation, attention, and adaptation.

We agree with the reviewer’s suggestion to explicitly frame the study within the context of the interplay between exploratory and defensive behaviors. To address this, we have added a dedicated paragraph to the Introduction section that highlights how animals balance these behavioral systems, especially in ambiguous or risky situations such as approaching the visual cliff. This paragraph also discusses how repeated exposures can lead to habituation, shifting the balance back toward exploration. We believe this addition better positions our study within broader frameworks of behavioral regulation, attention, and adaptation, aligning with the reviewer’s insightful comment. We have added the following to the introduction

“Importantly, animals constantly balance exploratory and defensive behaviors, especially in ambiguous or risky environments. When approaching a potentially threatening stimulus—such as the visually defined cliff—defensive responses tend to dominate. However, with repeated exposures, habituation may occur, shifting the balance back toward exploration. This dynamic interplay between behavioral systems is central to understanding how mice adapt their locomotor strategies in response to depth cues. By examining these state transitions, we can better interpret the observed behavioral signatures and situate them within broader frameworks of behavioral regulation, attention, and adaptation.”

Experimental Design

The circular open field design is a clear strength of the study. It elegantly avoids the corner biases that often confound results in square arenas and encourages more uniform spatial exploration. However, it would be helpful if the authors clarified whether this setup was controlled for potential confounds such as differences in lighting, floor reflections, or subtle asymmetries in visual patterns, which might influence movement patterns near the edges.

We acknowledge the reviewer’s point regarding potential confounds such as lighting, floor reflections, or subtle asymmetries in visual patterns in the circular open field setup. The square and circular setups were not conducted under strictly identical conditions, and therefore some potential for confounding exists. However, in our preliminary trials, we consistently observed corner-staying behavior in the square setup across various lighting conditions, while the circular setup consistently resulted in more uniform exploration regardless of the lighting conditions. We have added a note to the Methods section acknowledging this limitation and clarifying that the circular setup was found to reliably reduce corner-staying behavior across different experimental conditions.

“Preliminary trials were conducted using both square and circular arena setups under varying lighting conditions to assess the potential influence of lighting, floor reflections, and subtle visual asymmetries on locomotor behavior. We consistently observed corner-staying behavior in the square arena regardless of lighting conditions, while the circular arena consistently promoted more uniform spatial exploration. Although the square and circular setups were not strictly matched for lighting or other visual features, the robust reduction in corner-staying behavior in the circular arena was consistent across these different conditions, suggesting that the arena shape was the primary driver of the observed behavioral differences.”

Methodological Considerations

Regarding the analysis method, the use of Hidden Markov Models (HMMs) is one of the most distinctive features of the study. Yet the rationale for choosing this specific approach could be articulated more explicitly. Since the identification of behavioral patterns depends to a great extent on the analytical framework employed, it is important for the reader to understand why this model was favored—particularly whether it was selected a priori, or after comparing it to other options. Were alternative unsupervised learning methods (e.g., k-means clustering, PCA-based trajectory analysis, or duration-aware segmentation models) considered and rejected? What specific advantage does the HMM provide in terms of interpretability or fit to the biological data? Addressing these questions would help readers appreciate the methodological decision and interpret the resulting "three-state" model with the appropriate caution.

We appreciate the reviewer’s suggestion to clarify the rationale behind our choice of Hidden Markov Models (HMMs). We selected the Bayesian HMM approach a priori, based on its flexibility and its capacity to incorporate probabilistic state transitions and continuous temporal dependencies—features that are well-suited to capturing the dynamic behavioral patterns observed in our experiments. Unlike alternative unsupervised learning methods such as k-means clustering or PCA-based trajectory analysis, HMMs allow us to model the probabilistic switching between behavioral states while accounting for the sequential nature of locomotor data.

Moreover, the Bayesian framework offers the added advantage of incorporating prior knowledge, which is particularly useful in behavioral experiments where certain environmental factors (such as distance to the edge or center) are known to influence movement. By integrating these covariates and setting reasonable priors, we can create a model that aligns with known behavioral tendencies while maintaining enough flexibility to capture unexpected patterns.

While we did not directly compare alternative unsupervised methods in this study, we acknowledge that different analytical frameworks could yield different state definitions. We have added a note in the Methods and Discussion sections to clarify that the three-state model reflects the chosen modeling approach and should be interpreted as model-dependent, not as a definitive classification of all possible behaviors.

We added this to the Methods section

“We used a Bayesian Hidden Markov Model (HMM) to analyze locomotor behavior in the circular visual cliff arena. This approach was selected a priori because it captures the probabilistic switching between behavioral states and accounts for the sequential nature of locomotor data. The Bayesian framework allows us to incorporate reasonable prior knowledge about environmental factors (e.g., distance to the edge or center) that influence movement. By integrating these covariates and setting informative priors, we aimed to create a flexible model that reflects known behavioral tendencies while allowing for the detection of unexpected patterns. Although other unsupervised methods (e.g., k-means clustering, PCA-based trajectory analysis) exist, we chose the HMM approach because of its superior ability to handle temporal dependencies and interpretability in terms of state transitions. “

Relatedly, it is worth noting that the categorization into three discrete states—resting, exploring, navigating—is a product of the model architecture itself. While these categories are intuitive and align well with observed behavior, it would be valuable to acknowledge that other methods might yield different numbers or types of states. This does not diminish the value of the results, but it does suggest that the findings should be interpreted as model-dependent, and potentially open to alternative decompositions of behavior. A brief discussion on this point would demonstrate awareness of the limitations of statistical segmentation and encourage future comparative validation.

We agree with the reviewer’s point t

---

## [Decision Letter · Decision Letter 1]

16 Jul 2025

Hidden Markov Models Reveal Behavioral State Dynamics in Depth-Related Locomotion in Mice

PONE-D-25-07160R1

Dear Dr. Matsuyama,

We’re pleased to inform you that your manuscript has been judged scientifically suitable for publication and will be formally accepted for publication once it meets all outstanding technical requirements.

Kind regards,

Mario Treviño Villegas, Ph.D

Academic Editor

PLOS ONE

Additional Editor Comments (optional):

Reviewers' comments:

Reviewer's Responses to Questions

**Comments to the Author**

Reviewer #1: All comments have been addressed

Reviewer #2: All comments have been addressed

Reviewer #3: All comments have been addressed

2. Is the manuscript technically sound, and do the data support the conclusions?

Reviewer #1: Yes

Reviewer #2: Yes

Reviewer #3: Yes

3. Has the statistical analysis been performed appropriately and rigorously?

Reviewer #1: Yes

Reviewer #2: I Don't Know

Reviewer #3: Yes

4. Have the authors made all data underlying the findings in their manuscript fully available?

Reviewer #1: Yes

Reviewer #2: Yes

Reviewer #3: Yes

5. Is the manuscript presented in an intelligible fashion and written in standard English?

Reviewer #1: Yes

Reviewer #2: Yes

Reviewer #3: Yes

Reviewer #1: The requested comments have been addressed.

Please make greater use of recent sources. The way these findings can be translated into human studies and their neuroscientific applications to mitigate neurological issues arising from various disorders in the discussion section could be pivotal and delineate the future research trajectory in this field.

Reviewer #2: The authors have fully addressed all my comments. Good work. I have no more comments and wish to congrate the authors.

Reviewer #3: I am pleased with the revised version of the manuscript. The work is now in better shape and the authors have made an evident effort to address and amend the issues raised in the previous review.

Thank you very much for giving me the opportunity to review this interesting work.

**Do you want your identity to be public for this peer review?** For information about this choice, including consent withdrawal, please see our Privacy Policy

Reviewer #1: No

Reviewer #2: No

Reviewer #3: No

---

## [Editor Report · Acceptance letter]

PONE-D-25-07160R1

PLOS ONE

Dear Dr. Matsuyama,

I'm pleased to inform you that your manuscript has been deemed suitable for publication in PLOS ONE. Congratulations! Your manuscript is now being handed over to our production team.

Kind regards,

on behalf of

Dr. Mario Treviño Villegas

Academic Editor

PLOS ONE